



# PyGLDA: a fine-scale Python-based Global Land Data Assimilation system for integrating satellite gravity data into hydrological models

Fan Yang[1,2], Maike Schumacher[1], Leire Retegui-Schiettekatte[1], Albert I.J.M. van Dijk[3], and
Ehsan Forootan[1]

[1]Geodesy Group, Department of Sustainability and Planning, Aalborg University, Aalborg 9000, Denmark
[2]School of Physics, Huazhong University of Science and Technology, Wuhan 430074, China
[3]Fenner School of Environment & Society, College of Science, Australian National University, Canberra 2600, Australia

**Correspondence:** Fan Yang (fany@plan.aau.dk)

**Abstract.** Data Assimilation (DA) of time-variable satellite gravity observations, e.g., from the Gravity Recovery and Climate Experiment (GRACE), GRACE-Follow On (GRACE-FO) and future gravity missions, can be applied to constrain the vertical sum of water storage simulations of Global Hydrological Models (GHMs). However, the state-of-the-art DA of these measured Terrestrial Water Storage (TWS) changes into models is often performed regionally, and if globally, at low spatial resolution.

This choice is made to handle the considerably high computational demands of DA, and to avoid numerical problems, e.g., instabilities related to the inversion of covariance matrices. To fully exploit the potential of satellite gravity observations and the high spatial resolution of GHMs, we developed a Python-based open-source PyGLDA system that allows performing DA globally at a fine scale with high numerical efficiency. The main novelties of the system include (i) implementing a globe-scale patch-wise DA via domain localization and neighbouring-weighted global aggregation and (2) its great compatibility

between basin-scale and grid-scale DAs. This PyGLDA system represents a considerable functional advancement on previous implementations with wide and flexible options offered to allow for various user-specific studies. The modular structure of PyGLDA provides users with various possibilities to interact with (and add/remove) individual water storage compartments, change the representation of observations, and, therefore, the ability to choose different GHMs. In this paper, we present a full description of this system and its application for the Danube River Basin as a regional case study and through a global DA.

The DA demonstrations are performed using the monthly TWS fields of GRACE (2002-2010) and the W3RA water balance model at 0.1-degree/daily spatial-temporal resolution.

## 1 Introduction

Accurately monitoring the water cycle worldwide is essential for understanding the reservoir systems, climate change, water-related socioeconomic impacts and human management of water resources (Rodell et al., 2018). However, measuring large-

scale changes in Terrestrial Water Storage (TWS = a vertical summation of water in land including ice, snow, wetlands, lakes, rivers, soil moisture, and groundwater) and the individual water storage components via conventional methods has always been a challenge (Rodell and Reager, 2023). This is due to its considerable big dimension (covering the global land area = 150 million km$^2$) and the lack of measurements to monitor different hydrological variables. For example, access to in-situ





observations of these components (e.g., river discharge, soil moisture, and groundwater) is limited by spatial and temporal

coverage, and political interests. These data may also contain gaps and systematic errors, which make their application less
favorable in large-scale studies (Li et al., 2019).

Therefore, Global Hydrological Models (GHMs) of various structures are developed to simulate the distributed hydrological
response to weather and climate variations, as well as human water use. GHMs could be rainfall-runoff, e.g., DHI-GHM
(Murray et al., 2023), conceptual, e.g., WGHM (Döll et al., 2003), water balance, e.g., W3RA (van Dijk, 2010) and W3 (van

Dijk et al., 2018), as well as land surface models, e.g., NOAH-MP (Niu et al., 2011). One can see Sutanudjaja et al. (2018) for
a more extensive list of existing GHMs. These GHMs have been developed at different spatial resolutions, typically $0.5° \times 0.5°$
like WaterGap (Müller Schmied et al., 2021), or finer like $0.25° \times 0.25°$ for NOAH-MP, $0.1° \times 0.1°$ for DHI-GHM, $0.05° \times 0.05°$
for W3, and $0.01° \times 0.01°$ for the Hyper-resolution PCR-GLOBWB (Hoch et al., 2023). These models, however, might still
have difficulties in simulating all realistic water storage and water flux processes, which could be due to their oversimplification

of natural processes or the lack of constraints on the temporal variability of some components, e.g., groundwater and human
intervention. This issue is of relevance to all existing GHMs (Scanlon et al., 2018; Mehrnegar et al., 2020), though some might
work better than others in any particular river basin. To address this challenge, recent efforts have targeted to improve the
modelling of some compartments, e.g., groundwater representation in large-scale GHMs, to realistically simulate groundwater
levels and surface–groundwater interactions, whereas the imperfect modelling still remains an issue (see Niu et al., 2007; de

Graaf et al., 2017; Decharme et al., 2019). In addition, calibrating model parameters to better represent local water change
poses another major challenge, which requires innovative use of observations and calibration techniques (Schumacher et al.,
2018a).

Direct remote sensing observations of time-variable gravity changes have become available since 2002 by launching the
US/German Gravity Recovery and Climate Experiment (GRACE; Tapley et al., 2004) mission. By analysing these fields,

one can estimate TWS changes globally with reasonable accuracy (Landerer and Swenson, 2012). Over last two decades,
together with its follow-on mission (GRACE-FO; Landerer et al., 2020), these satellite-based TWS observations have helped
hydrologists and meteorologists to close the global water budget (e.g., Sheffield et al., 2009; Castle et al., 2016), to estimate
groundwater loss (e.g., Scanlon et al., 2012; Voss et al., 2013; Castellazzi et al., 2016), to study droughts and flood potentials
(e.g., Gouweleeuw et al., 2018; Forootan et al., 2019), to evaluate or validate models (e.g., Ngo-Duc et al., 2007; Eicker

et al., 2016; Mehrnegar et al., 2020), and to constrain or improve hydrological models (e.g., Güntner, 2008; Schumacher et al.,
2018a; Vishwakarma et al., 2021; Mehrnegar et al., 2023). Nevertheless, GRACE based TWS also faces the challenge of a
coarse spatial-temporal resolution ($\sim$500 km or $3° \times 3°$ at a monthly basis) due to the presence of its considerable correlated
noise (Kusche et al., 2009; Chen et al., 2022; Yang et al., 2024b).

To introduce realistic variability from observations into model simulations, the Data Assimilation (DA) technique has found

a great interest. Especially, the DA of satellite based TWS can improve temporal, horizontal and vertical disaggregation of the
'observed' TWS estimates (Zaitchik et al., 2008; Schumacher, 2016). TWS can constrain model based storage estimates, which
cannot be done with any other measuring techniques. A majority of DA studies in previous have confirmed its unique advantage
of improving TWS, as well as its compartments on regional/basin scale (Eicker et al., 2014; van Dijk et al., 2014; Schumacher





et al., 2016; Girotto et al., 2016; Khaki et al., 2017; Tian et al., 2017). In recent years, there is an emerging trend to implement
the DA on global scale to better reveal global water cycle (Li et al., 2019; Felsberg et al., 2021; Gerdener et al., 2023) and
its impact on climate change (Forootan et al., 2024), however, few of them can achieve high spatial resolution global DA of
GRACE-TWS and hydrological model. Li et al. (2019) claimed a $0.25°$ global DA, which is likely the highest resolution to the
best of our knowledge. While the value of high resolution (e.g., $0.1°$) hydrological model (as well as the meteorological forcing
field) has been acknowledged (Benedict et al., 2017; Sutanudjaja et al., 2018; Springer et al., 2019), running a high-resolution
global DA faces critical challenges that include: the weak numerical instability and limited computation resources, particularly
the latter of which shall lead to a complete failure of the traditional DA framework/system. Furthermore, unfortunately, none
of the previous publications that achieved a global DA of GRACE-TWS and model has made their software freely available,
and consequently, their technical implementations are not fully known to the community.

In this study, we present the first open-source, parallel, and Python based PyGLDA system (Yang, 2024) to address the
challenges of the High-Resolution Global Land DA (HRGLDA) for merging satellite based TWS (of GRACE, GRACE-FO,
and future gravity missions (Daras et al., 2023)) with GHMs. The software is an output of the DANSl-LSM project (https:
//aaugeodesy.com/dansk-lsm/), and will be extended in future for multi-sensor hydrological DA experiments. In PyGLDA, the
DA framework has been completely re-designed relative to previous studies to implement HRGLDA with 0.1 degree model at
daily time-step, efficiently and stably. PyGLDA includes new features that are detailed below:

– A novel unified framework to implement both the basin-scale and grid-scale DA, where the area to be computed is
defined as the 'basin' and the grid is treated as the 'sub-basin'. They can be introduced to PyGLDA by standard shape
files;

– A novel framework is introduced to seamlessly transit from regional DA to global DA by introducing domain localization
and neighbouring weighted aggregation algorithms;

– Computation of the spatial covariance of the observations, e.g., GRACE based TWS, is available for the first time to
evaluate full covariance matrices for sub-basins/grids. Accounting for the spatial correlations is allowed within the DA
of PyGLDA;

– Flexible options can be selected for the grid resolution of GHMs, where here a choice of $0.1°$ and $0.5°$ is available for
the employed global W3RA model (van Dijk, 2010);

– Flexible choices of the grid/sub-basin TWS observations to be assimilated into GHMs, e.g., from $1°$ to $5°$ with an
increment of $0.5°$;

– Flexible choices of perturbation strategies to generate ensembles, e.g., users can select which forcing data and which
model parameters to be perturbed following which noise distribution (Gaussian or Triangle distribution, as well as
additive or multiplicative noise);





– The Ensemble Kalman filter (EnKF similar to Schumacher, 2016), and an Ensemble Kalman Smoother (EnKS) that
     optimally desegregates monthly increments to daily (similar to Tian et al., 2017), are implemented as mergers.

The architecture of the code includes the features below:

– A flexible modular structure to de-couple PyGLDA into three individual modules: (1) GHM, (2) gravity data post-
  processing, and (3) mathematical DA integration. This choice is made to make it possible to easily develop/modify/re-
place any individual module;

– A high-level programming language (object-oriented Python) for easy comprehension of the code and to facilitate ex-
  tensibility. The Python translation of the W3RA model (van Dijk, 2010) is distributed;

– Easy/fast installation for cross-platform to the needs and capacity of users (e.g., Windows, Linux and parallel computa-
  tion at high performance computation clusters);

– Optimization by using a high-performance Numpy package as the basic data container to reach comparable numerical
  efficiency as C++, Fortran and Matlab;

– User-friendly interaction by (1) controlling/configuring PyGLDA with JSON setting file, where a wide options are avail-
  able for a different purpose; and (2) state-of-the-art data structure H5DF for reading and writing spatial-temporal data to
  allow for efficient management of data storage.

The paper is structured as follows. In Section 2, we describe the data used in the study. In Section 3, a detailed description
of GHM and the development of PyGLDA are described. Section 4 introduces a few application examples to showcase the ef-
fectiveness and flexibility of PyGLDA. Finally, in Section 5, discussions are provided, the study is concluded, and the outlooks
are demonstrated.

## 2   Data

### 2.1   Masks


Various data masks have been defined by PyGLDA to help manage the DA processes. A flexible combination of these masks
could allow for various user-specific studies as long as the masks are specified, whether the desired DA is basin-scale or grid-
scale and whether the DA is run globally or locally. Specifically, this is done by configuring the following five masks, regardless
of the spatial resolution: (1) land mask, which indicates grid points where the GHM is available; (2) data mask, where the
forcing fields and model parameter fields are available; (3) region mask, where one wishes to run the hydrological model; (4)
basin mask, where one wishes to run the DA; (5) observation mask, which can be sub-basins or grids with observations to be
integrated within the DA. In PyGLDA, there are default options for the first two masks; however, the last three masks must be
defined by the users via a standard shape file (.shp).



## 2.2 Meteorological forcing field

So far, only ERA5-land reanalysis (Hersbach et al., 2020) data has been integrated into PyGLDA. However, this does not mean that the implementation is restricted to this forcing field. Instead, one can easily modify the interface module to load other meteorological forcing data. In the current PyGLDA, the ERA5-land at a native (also its highest) resolution of $0.1°$ is required to run W3RA (as example), which includes the daily precipitation, the maximal/minimal temperature and the surface solar radiation. In particular, PyGLDA provides a function to download these forcing data from the ECMWF (European Centre for

Medium Range Weather Forecast) website automatically, which one can use to keep the forcing field up-to-date. In addition, an up(or down)-scaling of forcing fields will be done automatically if one wishes to run PyGLDA at a lower spatial resolution, e.g., $0.5°$, or higher, e.g., $0.05°$.

## 2.3 Parameters and Climatologies

The parameters and climate input data are originally distributed along with the GHM, i.e., W3RA, which have an original

resolution of $0.5°$. A finer resolution of $0.1°$ is provided in PyGLDA as default option. A detailed review of the definition of these parameters and climatology resources can refer to the technical document of W3RA (van Dijk, 2010).

## 2.4 GRACE Level-2 product and its variance-covariance

In PyGLDA, the monthly GRACE-based TWS is calculated from the standard level-2 products, which are gravity solutions expanded in terms of spherical harmonics coefficients up to a degree/order of 60 or 96. The level-2 products are often delivered

by three official data centers, i.e., from CSR (Center for Space Research from the University of Texas at Austin, Texas, USA), GFZ (GeoForschungsZentrum, Potsdam, Germany), and JPL (Jet Propulsion Laboratory, USA). In addition, the full variance-covariance information of L2 product is required to generate the error/perturbation of GRACE based TWS, however, this information is missing in current official L2 products. Therefore, we choose the ITSG-2018 temporal gravity products rather than the official ones (Kvas et al., 2019; Mayer-Gürr et al., 2018), because (i) ITSG-2018 is currently the only provider that

shares the normal equations of L2 products, from which one can obtain the full variance-covariance matrix; (ii) its variance-covariance matrix approximates the actual error structure very well (Kvas and Mayer-Gürr, 2019); (iii) a likely better quality (less noisy) than even the official ones could be expected (Meyer et al., 2019).

## 3 Model description

### 3.1 Structure

PyGLDA consists of three scientific modules: (1) GRACE postprocessing, (2) the W3RA GHM, and (3) the DA, which shall be individually elaborated on below. In addition, three supporting modules are involved: (4) a postprocessing module, which aims to formulate/save the DA output on desire and generate/save necessary visualization results, (5) an auxiliary module for dealing with general operations such as forcing data downloading, downscaling/upscaling, visualization, statistical computation etc.,

**Figure 1.** An overview of PyGLDA structure that mainly consists of three modules: (a) GRACE processing, (b) W3RA GHM (van Dijk, 2010) and (c) Data Assimilation (DA).

and (6) a workflow control module for integrating the necessary sub-modules together as a chain to accomplish a specific study.
Figure 1 provides an overview of the structure of PyGLDA platform.

**GRACE processing**. The goal of this module is to accomplish the *conversion*, the *correction* and the *propagation* of GRACE level-2 data/error. GRACE level-2 product, by convention, is distributed to the public in terms of geopotential spherical harmonic coefficients (SHCs). Therefore, it is our responsibility to convert these SHCs to a desired physical variable for a specific study, for instance, gridded TWS anomalies to be used as observation in DA, which is called the 'conversion' via a thin-layer

harmonic synthesis (Wahr et al., 1998). In addition, a series of 'corrections' is recommended to ensure a clean signal extraction. This includes (but not limited to) low-degree coefficient replacement (e.g., Loomis et al., 2020), de-noising (e.g., Swenson and





Wahr, 2006; Kusche, 2007; Klees et al., 2008; Horvath et al., 2018), spatial leakage correction (e.g., Chen et al., 2015; Vishwakarma et al., 2018), removing the GIA (Glacial Isostatic Adjustment, e.g., Richard Peltier et al., 2018), Ellipsoidal correction (e.g., Yang et al., 2022) and so on. Each *correction* step is associated with numerous strategies, of which the most popular ones

are selected and implemented by PyGLDA, see Table 1. Moreover, quantifying uncertainty for GRACE-based TWS is essential for the DA to reach an optimal compromise of the water balance between observations and model simulations. This is done by a *propagation* of the level-2 full variance-covariance matrix (in terms of SHCs) into the gridded (or the basin-averaged) TWS (Wahr et al., 2006; Boergens et al., 2022). In PyGLDA, we implement a Monte-Carlo method to acquire the final variance-covariance of TWS, where the covariance is rarely available before this study (Yang et al., 2024a). In addition, considering the

independence of GRACE processing, we integrate all GRACE processing-related jobs into a stand-alone toolkit, which can be coupled with PyGLDA seamlessly (Shuhao, 2024).

**W3RA GHM**. W3RA is an open-source and global realization of AWRA-L (Australian Water Resources Assessment system, version 0.5) hydrological model (van Dijk, 2010). It is a one-dimensional, grid-based water balance model that has a lumped representation of the water balance of the soil, groundwater and surface water stores, where available, simple and

established equations are used to describe processes determining the radiation, energy and water balance. Minimum meteorological inputs are daily gridded estimates of precipitation, incoming short-wave radiation, and daytime temperature (currently estimated from minimum and maximum temperature). Ideally the model can be applied at any resolution; the implementation of public W3RA is at $0.5°$, but we modify it to $0.1°$ to be compatible with the resolution of available meteorological data (Mehrnegar et al., 2021). An important caveat for the current model version is the assumption to ignore lateral redistribution

between grid cells, which is less reliable when interpreting the results for areas that may be subject to irrigation, inundation or groundwater inflows (van Dijk, 2010), and this motivates the refinement by incorporating GRACE-TWS with DA. Original W3RA was released in an open-source Matlab script, but it has been translated to Python and integrated into PyGLDA, where intensive code validations across various combinations of areas/resolution/forcing have been made to verify the same precision and comparable numerical efficiency between before/after translation (based on 0.5 degree solution). The TWS simulations

of W3RA is compared in Mehrnegar et al. (2020) against GRACE TWS and with other commonly used GHMs, which were forced by the same climate data within the eartH2Observe project (Schellekens et al., 2017). Their comparison indicated that the skills of W3RA in resembling the global water storage variability are similar to the existing models.

**Data Assimilation (DA)**. This module consists of four main steps, which are *perturbation*, *initialization*, *forecast*, and *analysis*. As Ensemble Kalman Filter (EnKF) or Ensemble Kalman Smoother (EnKS) are selected as our merger of DA, a proper

*perturbation* of both model and observations has to be performed. In PyGLDA, the *perturbation* of the model (specifically, the model parameters and forcing fields) can be generated by either additive or multiplicative way, adhering to either Guassian or Triangle distribution (Schumacher, 2016). On the contrary, the *perturbation* of GRACE observations is generated from the full variance-covariance information. Be aware that the *perturbation* of the initial field (state vector) is currently not considered in PyGLDA due to its limited impact on the land water DA (Reichle and Koster, 2003). In the next step, an *initialization*, i.e., a

spin-up of the error-free W3RA GHM (given, for example, two years at least) to acquire a reliable initial field. Subsequently, in the *forecast* step, the state of the model is continuously evolving on a daily basis until being informed to have an observation





**Table 1.** An overview of available options of *correction* in PyGLDA's GRACE postprocessing toolkit.

| Correction | Options | Remark | Reference |
|---|---|---|---|
| Low degrees replacement | Degree-1 | $[C_{1,0}, C_{1,1}, S_{1,1}]$ by model | Swenson et al. (2008) |
| | $C_{2,0}$ | by TN-11[1] file as of December 2019. | Loomis et al. (2020) |
| | $C_{3,0}$ | by TN-14 | Loomis et al. (2020) |
| De-correlation | Sliding window fitting | Empirical selection of window length | Swenson and Wahr (2006) |
| | P$n$M$m$ | A polynomial fitting for degree $n$ and order $m$ | Chen et al. (2006) |
| Averaging filter | DDK[2] | DDK1 - DDK8 | Kusche (2007) |
| | Gaussian | with a radius $r$, $200 < r < 1000$ (km) | Jekeli (1981) |
| | Non-isotropic Gaussian | $r_1$-radius of degree, $r_2$-radius of order, $m_0$-truncated order | Han et al. (2005) |
| | Fan | $r_1$-radius of degree, $r_2$-radius of order | Zhang et al. (2009) |
| Leakage correction | Forward modeling | Iteration times $n$, acceleration factor $s$ | Chen et al. (2015) |
| | Classical[3] | $\hat{f}_c = \overline{f}_c - l_c$ | Wahr et al. (1998) |
| | Additive[4] | $\hat{f}_c = \overline{f}_c - l_c^m + b_c^m$ | Klees et al. (2007) |
| | Multiplicative[5] | $\hat{f}_c = k(\overline{f}_c - l_c^m)$ | Longuevergne et al. (2010) |
| | Scaling | $\hat{f}_c = k^m \overline{f}_c$, basinal or gridded | Landerer and Swenson (2012) |
| | Data-driven[6] | $\hat{f}_c = \overline{f}_c - \overline{\delta F}_c - l_c$ | Vishwakarma et al. (2018) |
| GIA correction | ICE6G-D (ICE series) | Removal in spectral domain or in spatial domain | Richard Peltier et al. (2018) |
| | Geruo2013 | | A et al. (2013) |
| | Caron2018 | | Caron et al. (2018) |

[1] The Technical Note (TN-XX) files are provided along with the official GRACE products.

[2] DDK is actually a de-correlation and averaging filter (Kusche et al., 2009).

[3] where $\hat{f}_c$, $\overline{f}_c$ denote the signal after and before correction; and $l_c$ indicates the spatial 'leakage'.

[4] where $b_c$ indicates the 'bias'; and the superscript $m$ indicates that a hydrological model is required.

[5] where $k$ is the derived scaling factor.

[6] where $\overline{\delta F}_c$ indicates the deviation integral.

available. Finally, in the *analysis* step the observations and model are combined together to renew the states of the past month following the general principle of EnKF or EnKS, see more details in Appendix A.





## 3.2 DA workflow and parallel computation

The state-of-art-of sequential DA is a common practice in hydrological DA applications, where a Kalman-like filter has found a great interest due to its relatively straight forward matrix implementation (Schumacher, 2016). A classical Kalman filter assumes that all probability distributions involved in the dynamic process are Gaussian and provides algebraic formulations for the evolution of the mean and the covariance matrices according to the Bayesian update, provided the system is linear. However, maintaining the covariance matrix is not feasible for high-dimensional systems. For this reason, EnKF was developed, which

represents the distribution of the states using a collection of state vectors called an ensemble and replaces the covariance matrix with the sample covariance computed from the ensemble (Evensen et al., 2009). Therefore, the EnKF is a Monte Carlo implementation of the Bayesian update, where a number of samples (ensembles) are required for implementation. Since there is no agreement on how many ensembles have to be used to ensure an accurate covariance approximation, PyGLDA allows for an arbitrary number of ensembles to the user's need and capacity. Configuring the number of ensembles can be easily done via

the GHM setting file, which is formulated in a standard .json file, see Appendix B.

Given the number of ensembles required, the processing (i.e., the perturbation, the model forecast, the state analysis and the data management) of each individual ensemble would be completely independent of each other. This property makes it possible to parallelize the computation to fully exploit the CPU resources, where each thread is assigned to be in charge of one ensemble. In PyGLDA, the parallelization is implemented by leveraging the MPI (Message Passing Interface), which is a

standardized and portable message-passing system designed to function on a wide variety of parallel computers. Specifically, the package MPI4PY provides MPI bindings for Python that allow exploiting multiple processors and even multiple nodes on clusters.

Figure 2 provides a schematic diagram to illustrate how the parallel DA works in PyGLDA, where an ensemble of two members has been shown as an example. In practice, one use more ensemble members, for example 30 in (Schumacher et al.,

2016, 2018a). The general DA workflow, as defined in PyGLDA, is divided into three stages: (1) a *single-run*, which is the error-free model run, (2) an *open loop*, which consists of various runs with perturbed forcing and/or parameters, and (3) the *data assimilation* step that merges ensemble members with the ensemble of observations. At the *single-run* stage, the hydrological model spins up for two years (that can be longer) so that the GHM reaches its balance. This acquires reliable initial state vectors for the later use (Viney et al., 2015). Consequently, the *single-run* indicates a perturbation-free run of the original

model, where only one single thread is required. At the *open loop* stage, the ensembles are running simultaneously (with only model forecasts) for the specified time period (e.g., 2002-2020). In this situation, PyGLDA launches multiple threads, each handling one ensemble member. Each is initialized with the state derived from the *single-run* stage and uses its own perturbed parameters and forcing fields. Subsequently, for each ensemble member, the long-term mean of the state vector is calculated and subtracted from the GRACE observation. At the *assimilation* stage, multiple treads are launched again, whereas only one

of them is meant to deal with the Kalman analysis step (see Appendix A for more details) when an observation is available. In summary, a complete DA job will go through these three stages to arrive at the final assimilated results, where PyGLDA automatically manages the threading pool to distribute proper threads for each stage.





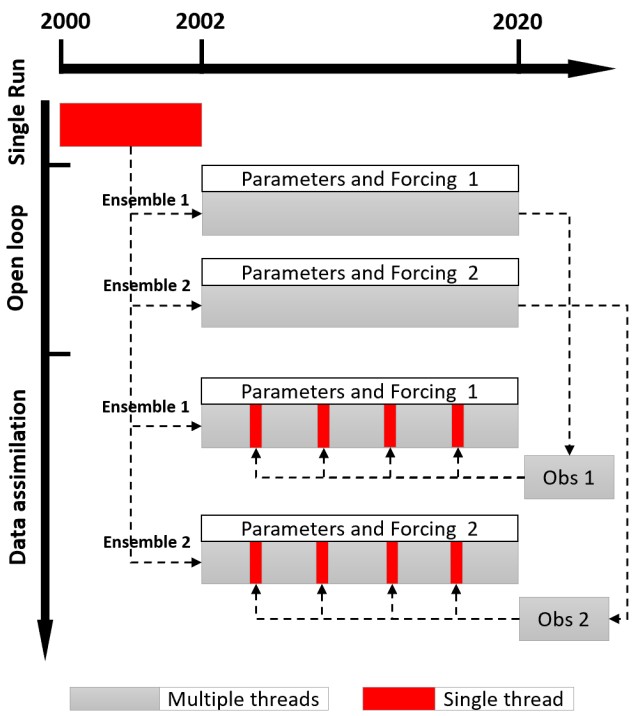

**Figure 2.** A schematic diagram to manage the threading pool of the DA workflow in PyGLDA, where an ensemble of two members for 2002-2020 is shown as an example.

## 3.3 GLDA configuration

### 3.3.1 Patch-wise domain localization

Running a high-resolution GHM, particularly with an ensemble of over dozens of members, is rather challenging for general computation platforms, where the resources are often limited (Reichle and Koster, 2003). On the other hand, from the perspective of DA, the model's state vector that spans at a global scale can easily introduce spurious and unexpected long-range correlations if the size of the ensemble is too small (Forman and Reichle, 2013; Khaki et al., 2017). Such artificial covariance often degrades DA computation, which behaves as the instability of covariance inverse during the Bayesian update step of

EnKF, and this may even lead to a divergence of the filter. In summary, compared to the regional DA, the traditional GLDA (Global Land DA) has to face additional difficulties of less computational affordability and less numerical robustness, which challenge the implementation of GLDA in practice. Consequently, far less GLDA of GRACE and hydrological model has been attempted in the literature than the regional DA.

     To address this challenge, it is desirable to restrict the area where covariance matrices are calculated, e.g., over a given

tile. This introduces a 'domain' specific localization between tiles that regularizes the whole covariance matrix. By this, our hypothesis is that considerable correlations do not exist beyond the selected area. On one hand, by assuming so, one can



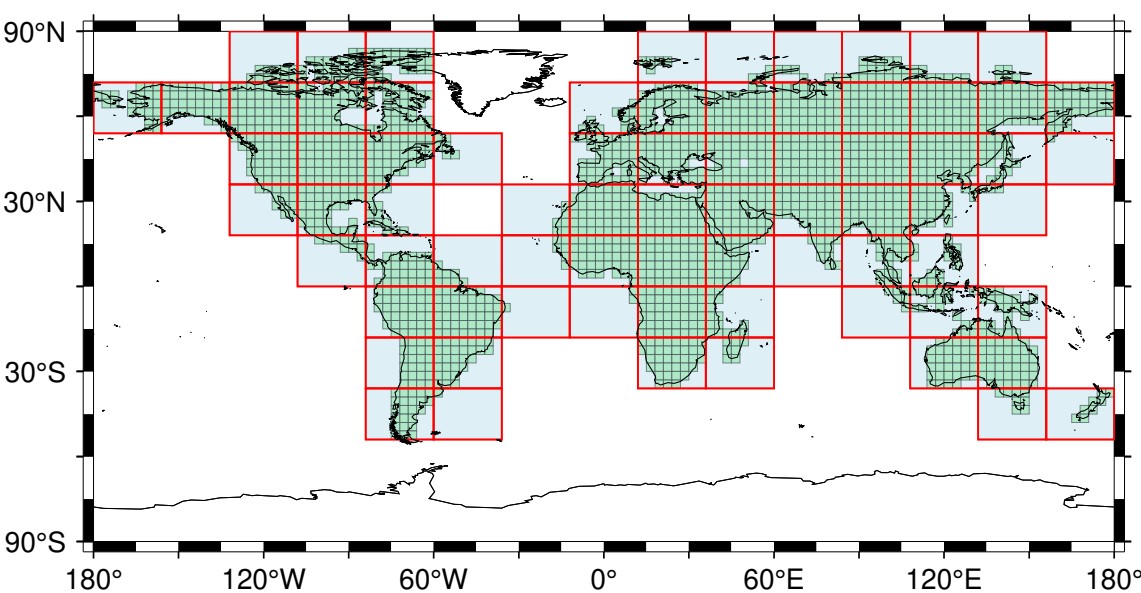

**Figure 3.** Configuration of the patch-wise domain localization, where each tile (denoted as the rectangle filled by the light blue) represents the 'domain' to perform DA, individually. The unit box in black, contained in each tile, represents the smallest area to average GRACE observations to be assimilated.

effectively reduce the spurious long-range correlation and stabilize the filter. On the other hand, if such a hypothesis is valid for both the model and GRACE observations, the DA implementation within the area can be considered independent from beyond. In other words, this means that GLDA can be divided into various patches/tiles, each performing its own DA, given that the area

of the tile is large enough to ensure independence from its neighbours. In Figure 3, an example of this GLDA configuration (the domain localization) is illustrated. In this example, the smallest unit box is defined as $3° \times 3°$, where $0.5° \times 0.5°$ GRACE-TWS is integrated and averaged. This is to fully exploit the native spatial resolution of GRACE (Forman and Reichle, 2013; Li et al., 2019). For future gravity missions, such as MAGIC and NGGM (Daras et al., 2023) the grid resolution can be smaller.

In addition, the 'domain', where individual DA is performed, is selected as $18° \times 24°$ as indicated by the red rectangular box

in Figure 3. The size of the domain is indeed flexible in PyGLDA, and the one in Figure 3 is an example but can be considered as a recommended option for GRACE global DA. In the end, by removing meaningless domains that have no common area with the land mask. In total, there are 74 valid domains that shall participate in the DA computations (excluding the glacier areas). In practice, one can change all the configuration as desired, e.g., to define the unit box as $4°$ or $5°$ and change the domain to arbitrary size as long as the mask is defined. By this, the numerical difficulties of instability and limited computation resources

brought by the large-scale DA could be overcome. In particular, parallel computation of patch-wise DA can be facilitated to allow GLDA to run on a small cluster or even on a personal computer efficiently.



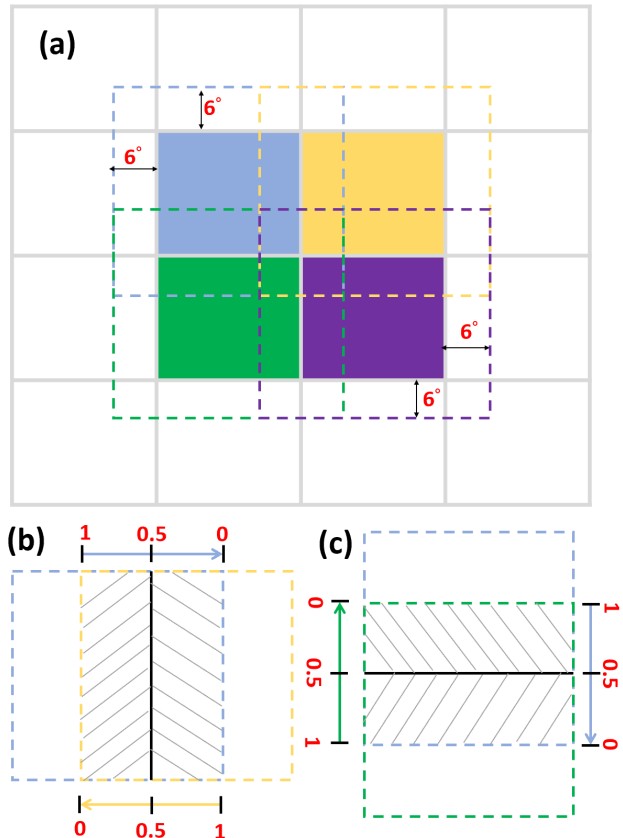

**Figure 4.** An illustration of the proposed global aggregation strategy. In Figure 4(a), the box in grey replicates the tile/patch of Figure 3. Four adjacent tiles (in blue/green/yellow/purple) are selected as examples to show the extension of the zone (in dashed line, e.g., 6°). A zoom-in is demonstrated in Figure 4(b-c), where the shaded area indicates the overlap of two extended tiles, and the red number indicates the weight assigned to each tile for the averaging over the overlapping areas.

### 3.3.2 Weighted global aggregation

In practice, the hypothesis of independence between adjacent domain/tiles is a strong assumption. While it is reasonable to assume that the correlation between the center of each tile and points outside of the tile is zero (it is long-range when the domain/tile is big enough), the correlations between points that are close to edges are usually not zero (Reichle and Koster, 2003). Especially since two tiles are cut off from the middle as indicated by Figure 3, neglecting correlations between points across the border of tiles would lead to an unexpected 'edge' effect after the global DA. Such an edge effect would appear as a clearly distinguished line between neighbouring tiles, which would indeed make no physical sense. Instead, the desired transition between neighbouring tiles is supposed to be smooth.

Since it has been recognized that the 'edge' effect is caused by neglecting the correlation of area bordering adjacent tiles, we propose to extend the size of the tile to allow for an overlap of adjacent tiles, and such an overlapping can be regarded





as the transition area. In this way, DA results of adjacent tiles over the transition area could be combined together to account for the correlation that was previously ignored. In particular, a weighted averaging of adjacent patches is suggested to ensure a smooth combination/transition, so that one can expect to well alleviate the 'edge' effect. Figure 4 illustrates the schematic

diagram of the proposed strategy, which mainly consists of two steps: (1) zone extension and (2) weighted averaging. In step-1, as indicated by Figure 4(a), the tile/domain is defined following that of Figure 3, whereas an extension (e.g., 6°) is applied to each tile (in dashed line), where the DA is supposed to be performed. Such an extension slightly increases computation effort, but it is worthwhile and still computationally feasible, as verified by cursory testing. When the patch-wise DA is finished, an aggregation of these patches as a whole has to be performed, which comes to Step 2 in Figure 4(b-c). In this step, there are

likely two types of combination, i.e., an aggregation along the horizontal profile as indicated by Figure 4(b), and that along the vertical profile as indicated by Figure 4(c). Nevertheless, the operation is common, i.e., DA results of adjacent tiles are averaged over the shaded area where they overlap each other, each being assigned with a weight proportional to the distance from its outermost border, e.g., $f^a(x) = \sum_{i=1}^{2} w^i(|x - b^i|)f^i(x)$, and where $f^a$ denotes the averaged results from two patches ($f^1$ and $f^2$), $x$ denotes the certain point inside the overlapping area, $b^i$ denotes the boundary of $i^{th}$ patch, and $w$ indicates

the weighted function related to the distance $|x - b^i|$. For example, in Figure 4(b), the weight of the blue area is assumed to be $1 \rightarrow 0$ from left to right, and vice versa, the weight of the yellow area is $0 \rightarrow 1$. The inverse-distance weight is reasonable considering the fact that correlation often decays with the distance (Reichle and Koster, 2003).

## 4 Application

### 4.1 Basin-scale DA for Danube

The Danube River Basin (DRB) is Europe's second largest river basin, with a total area of 801,463 km². It is the world's most international river basin, flowing through the territory of 19 countries. Understanding how the terrestrial water changes over DRB is important for the water management of these countries. Therefore, we select DRB to perform a case study to illustrate PyGLDA's ability of a regional DA case study. In this experiment, as basin-scale DA is widely used by previous studies, we also configure the experiment as a basin-scale DA. A gridded DA over DRB will be demonstrated in the next section.

To be able to run basin-scale DA, the domain of the DRB, as well as its subbasins, need to be defined. By convention, the DRB can - based on its gradients - be divided into three sub-regions: the upper basin, the middle basin, and the lower basin (Stagl and Hattermann, 2015). The Upper Basin (UB) extends from the source of the Danube in Germany to Bratislava in Slovakia. The Middle Basin (MB) is the largest of the three sub-regions, extending from Bratislava to the dams of the Iron Gate Gorge on the border between Serbia and Romania. The lowlands, plateaus and mountains of Romania and Bulgaria form

the Lower Basin (LB) of the River Danube. Please see also our definition of DRB and its subbasins (in dark blue) in Figure 5. As mentioned before, the 'subbasin' is defined to be the area where GRACE data shall be averaged as observation.

As outlined in Figure 1, the module of GRACE processing is first launched to generate observations to be assimilated and a summary of its configuration is reported in Table 2. In addition, a Monte Carlo propagation of the full variance-covariance from spherical harmonic coefficients to the TWS grids can be easily obtained via Yang et al. (2024a), and in this way, the variance-



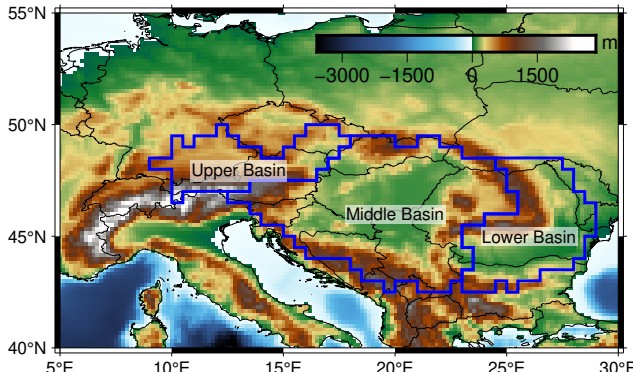

**Figure 5.** The Danube River Basin (DRB) and its three major subbasins. The basin's topography is shown as background picture.

**Table 2.** GRACE processing choice for the case study of DRB.

| Step | Option |
|---|---|
| Low-degree replacement | degree-1 terms and $C_{2,0}, C_{3,0}$ |
| Filtering | DDK3 |
| GIA correction | ICE6G-D |
| De-mean | temporal mean of 2002-2010 |

covariance of subbasins can be obtained as well. It is worth mentioning that, in previous studies the correlation between subbasins was often assumed to be zero; however, in our study, it is found that there is likely evident correlation between subbasins, for example, the correlation coefficient between UB and LB reaches -0.50 on 2009-01. In this sense, accounting for a full variance-covariance into DA should be more reasonable. And the temporal variability in the variance-covariance should be taken into account as well since it has reflected the height adjustment of GRACE orbit that shall significantly change the

structure of derived variance-covariance, particularly GRACE's quality degrades a lot around the end of the mission (Boergens et al., 2022; Yang et al., 2024a).

Empirically, an ensemble size of 30 was considered a reasonable trade-off between computational cost, storage capacity and representative error statistics (Schumacher et al., 2018b). Nevertheless, the option of ensemble size is flexible in PyGLDA, which should be defined by users on their own. However, the threads initialized by MPI must be consistent with the defined

ensemble size, otherwise an error would be reported for insufficient threads handling the ensemble. The model ensemble can be generated by introducing perturbation into the meteorological forcing fields, climatological fields, parameters fields, and even the initial fields. In this case study, the perturbation of forcing fields and parameter fields are taken into account. Specifically, all these fields were perturbed using random Monte Carlo sampling from triangular probability density functions by introducing



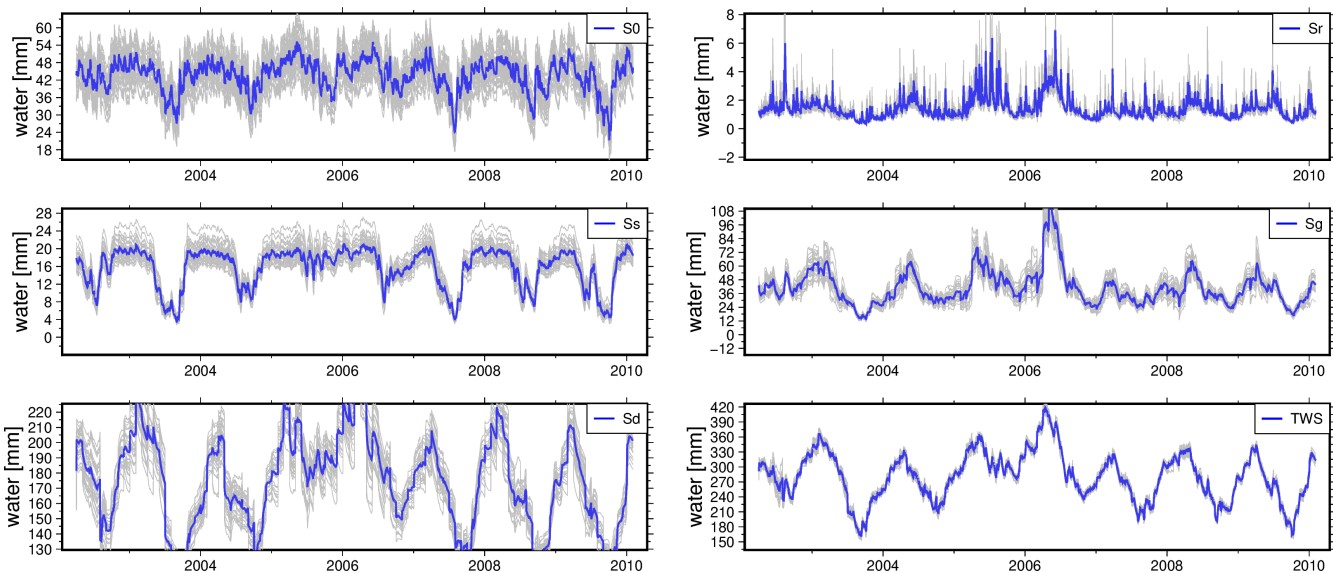

**Figure 6.** An overview of the mean (in blue) of the ensemble (shaded) estimates after Basin-scale DA experiment over DRB is illustrated. Plots include the TWS estimate and its major vertical compartments.

a multiplicative error of 30%. In the current version of PyGLDA, the perturbation is assumed to be spatially correlated but

temporally uncorrelated, which should be developed in future to allow for arbitrary local scale of spatial-temporal correlation in generating perturbations.

Following the strategy described before, we perform the DA over DRB from years 2002 to 2010 and collect the ensemble in Figure 6. Therefore, by convention, the average of ensembles can be assumed to be the assimilated result, and moreover, the spread of the ensemble should represent the uncertainty, while it is suspicious how reliable and to what extent the realistic

uncertainty can be represented (Gerdener et al., 2023). The variability of TWS and its five major components are illustrated in Figure 6, from which we estimate the variance of each component and understand how DA distributes the increment to different components. The selected five components include the surface water ($Sr$), the topsoil water ($S0$), the shallow soil water ($Ss$), the deep soil water ($Sd$) and the groundwater ($Sg$), which altogether constitute the state vector of W3RA model. If sorted by the variance, the contribution of GRACE TWS to the model state is declining from $Sg$, $Sd$ (comparable to $Sg$) to $S0$,

$Ss$, $Sr$, accordingly. In other words, the contribution of TWS assimilation mostly goes into the groundwater and the deep soil water components. On the contrary, the other components mostly absorb the high-frequency contribution from GRACE.

Furthermore, the basin-averaged TWS of DRB and its three major subbasins as defined in Figure 5 have been shown in Figure 7, where the results of OL (Open-Loop, indicating pure model prediction), the DA, and the GRACE observations are compared. By contrasting the GRACE (black dot) with model OL (shaded line), one can see that more dynamic is preserved

in the model, but a larger magnitude is preserved in the observations, indicating the need of DA to compromise model and observations. By contrasting the DA (in green) to GRACE/OL, one can see that the magnitude of the TWS is adjusted closer

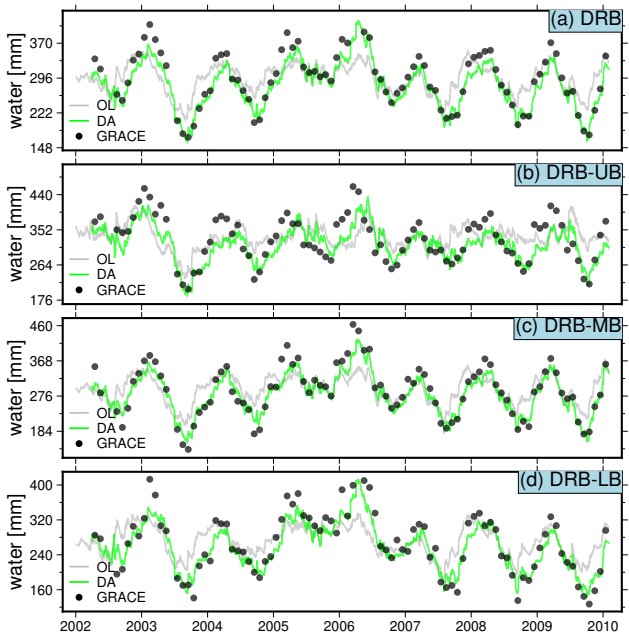

**Figure 7.** An overview of the basin averaged TWS of DRB and its three subbasins (LB, MB, and UB) for OL, DA and GRACE observations.

to the observation as expected, and in the meantime, the dynamic of the model is well preserved as well. In addition, from Figure 7(b) we see an evident phase shift between OL and GRACE due to likely a lag of stream flow, but in the DA result the phase shift has been compromised, which, however, is hard to justify without in-situ validations.

At last, we extract the seasonality of these signals (OL, DA and GRACE) and demonstrate their 2-D spatial maps in Figure 8 and in Figure 9. Please be aware that the original spatial resolution of OL and DA is $0.1°$, but an up-scaling (by aggregating/averaging adjacent cells into a bigger cell) has been applied to enhance comparisons with GRACE observations that are computed at a square cell of $0.5°$, see Figure 8(a-b) and Figure 9(a-b). In addition, to obtain a better visual comparison, a 2-D spline smoothing is applied to all these results, see Figure 8(d-f) and Figure 9(d-f). All these operations have been implemented

and integrated in PyGLDA. The secular trend in Figure 8 indicates that DA resembles the OL (model prediction) very well at a majority of DRB, except for its subbasin UB, where DA result is closer to GRACE. Based on Figure 8(d-f), the spatial correlation between the DA and OL is 0.74 (for UB), 0.98 (for MB) and 0.89 (for LB); the spatial correlation between DA and GRACE is 0.79 (for UB), 0.35 (for MB) and 0.75 (for LB). The low correlation of MB between DA and GRACE indicates that DA relies more on the model when the model and observation has a large discrepancy. Likewise, by looking into the annual

amplitude (which dominates the signal over DRB), it is found from Figure 9 that DA perfectly replicates the spatial pattern of OL, e.g., their spatial correlation reaches up to 0.92; but meanwhile, its spatial magnitude has been tuned much (almost twice bigger than OL) toward GRACE observations. In this sense, PyGLDA has fulfilled its goal of balancing the model and observations to reach the statistically optimal combination. Detailed validations will be done in follow-up studies.



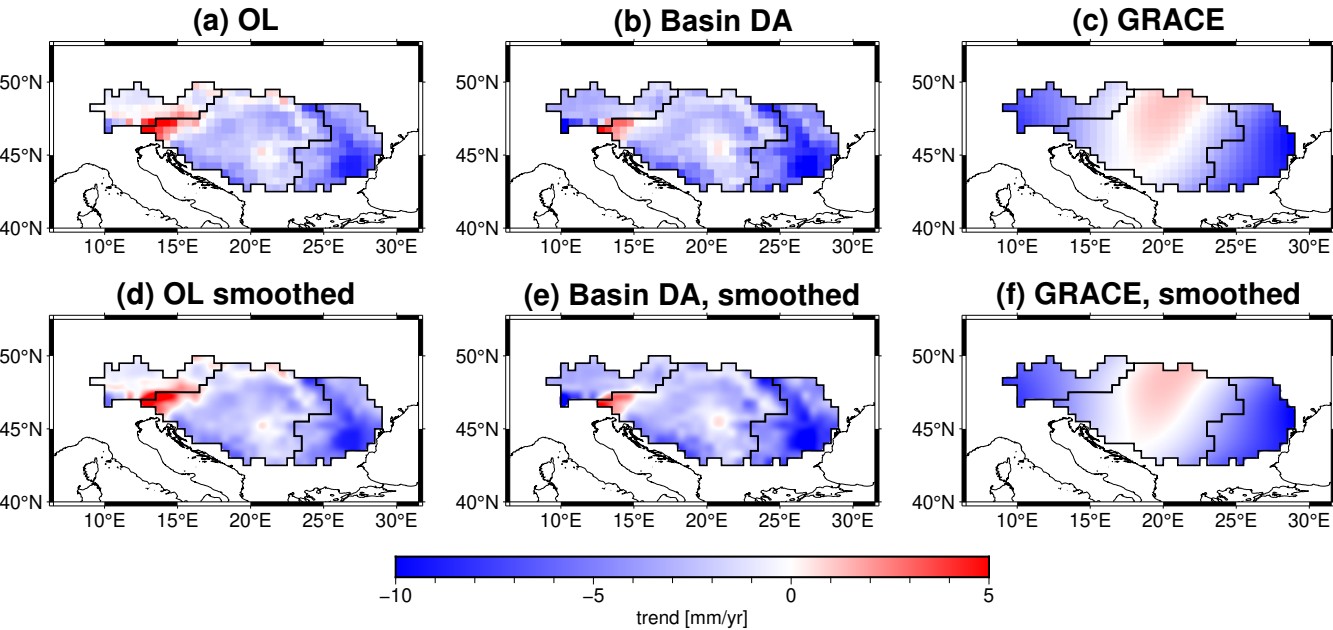

**Figure 8.** The secular trend of TWS over 2002-2010 for DRB, where the OL, DA and GRACE results are listed at (a-c), and a smoothed version is respectively shown at (d-f) for a better visualization.

## 4.2 Grid-scale DA within DRB

In recent years it has been a trend to apply grid-scale GRACE data, e.g., the $3° \times 3°$ Mascon solution from JPL, to perform the DA with a high-resolution hydrological model, since this way the potential of GRACE is likely better exploited (Girotto et al., 2016; Khaki et al., 2017; Li et al., 2019; Gerdener et al., 2023). In this section, we shall perform another DA experiment over DRB again, but the strategy changes from the basin scale (as described in the previous section) to the grid-scale. It is worth mentioning that PyGLDA launches an instance of DA experiment as long as the shape file of the target region (a basin as well

as its subbasin) is defined, regardless of whether the intended DA is basin-scale or grid-scale. In other words, as long as the shape file is defined for the 'grid', the PyGLDA is able to accept the shape file and run the gridded DA on demand without the need to change any structure of the code.

In this experiment, we follow the most widely used choice of grid resolution (i.e., $3° \times 3°$ for GRACE), and they are defined to completely cover DRB, see Figure 10 the index and the shape of each cell. In the end, the whole DRB can be divided into

19 cells, and each square cell is intercepted with the border of the DRB to form a new cell. As one can see from Figure 10, the size/shape of each cell is not uniform, and some of them even appear odd, such as the 7th cell. An optimal grid generation for basins of various shapes can be considered in future. Following the configuration of Figure 10, the GRACE TWS (originally $0.5°$) is averaged over each cell and generates the cell-wise observation. Likewise, the spatial covariance (correlation) of these cells is investigated in Figure 11, where one can see that (i) the correlation is strong between cells, e.g., some even goes beyond



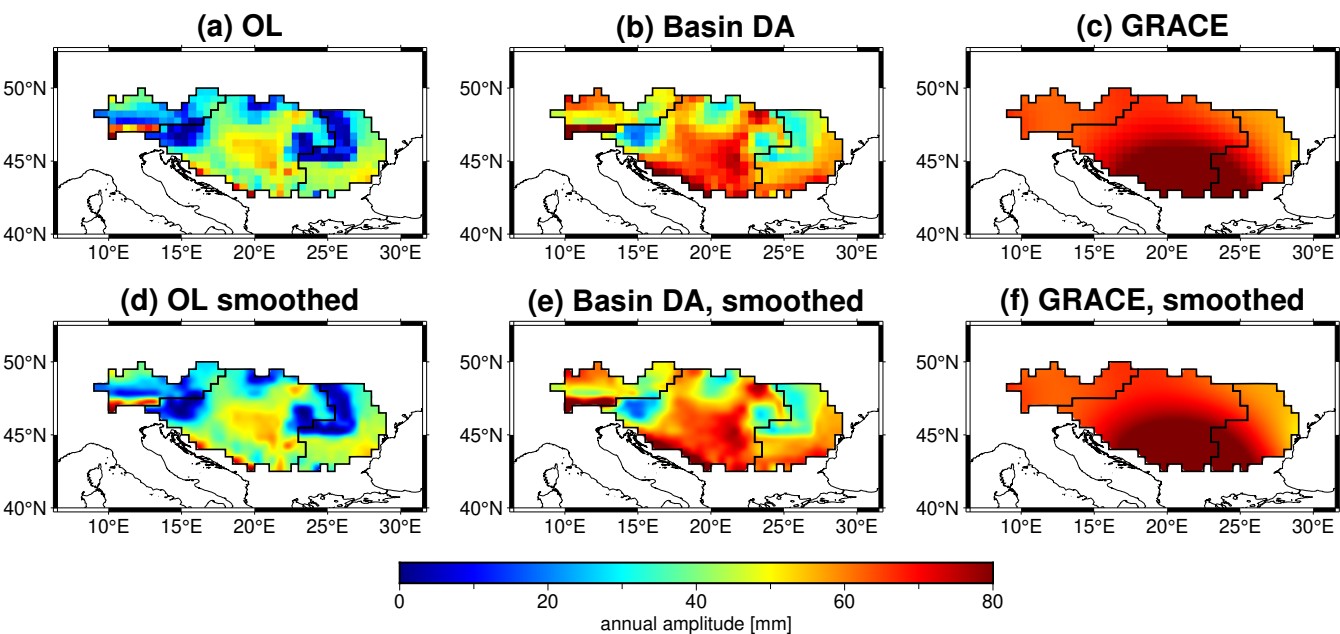

**Figure 9.** An overview of the annual amplitude of TWS within DRB over 2002-2010, where the OL, DA and GRACE results are shown in (a-c), and a smoothed version is respectively shown at (d-f) for a better visualization.

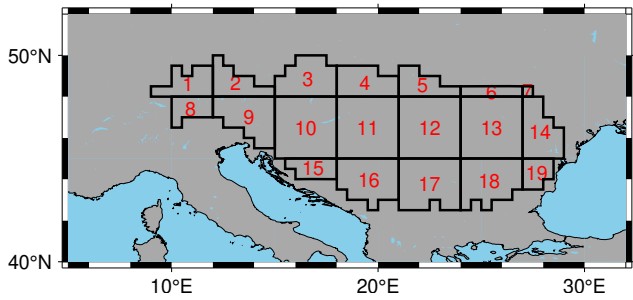

**Figure 10.** An overview of the $3° \times 3°$ cell definition for DRB, where the identification number (ID) of each cell is highlighted in red.

0.9, and (ii) the evident negative correlation appears, manifesting as strong oscillating side-lobe because of imperfect decorrelation filtering applied to GRACE. Based on these findings, it seems assuming a non-correlated perturbation for GRACE is not reasonable, and the traditional variance localization (assuming a decaying covariance with range) might not work as well. Therefore, PyGLDA always prefers the full variance-covariance matrix of GRACE rather than the diagonal entries or localized covariance estimates.

Keeping the rest of the configuration unchanged from the previous section (e.g., the strategy of GRACE post-processing as well as the perturbation etc.), we performed so-called Gridded DA and present the ensemble result in Figure 12, where, for brevity, only the result of whole DRB is demonstrated. Comparing Figure 12 to Figure 7(a), we found that the basin-

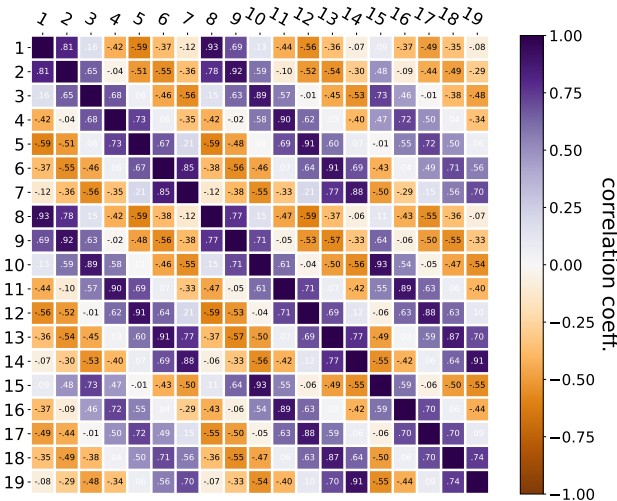

**Figure 11.** Spatial correlation of GRACE observations for cells within DRB. The label of X(Y) axis denotes the cell ID as indicated by Figure 10.

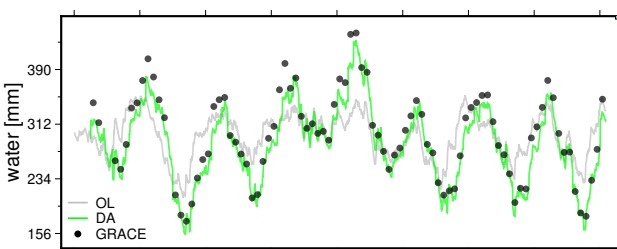

**Figure 12.** An overview of the basin averaged TWS of DRB for OL, grid-scale DA and from GRACE observations

averaged TWS estimates of these two scenarios are generally similar, but some minor differences can be detected. The temporal correlation of these two curves is as high as 0.99. This indicates that the statistics of Gridded DA over the basin-scale area would not change much relative to that of Basin DA. However, this does not take away the value of Gridded DA, which expects to take its advantage at a finer scale, see the 2-D map in what follows. Instead, such consistency of large scale is able to prove that our implementation of the 'Gridded DA' and the 'Basin DA' is correct.

Comparisons are performed between the 2-D spatial maps of Figure 13 and Figure 14, in terms of the trend and the annual amplitude at a spatial resolution of $0.5°$ for 2002-2010. Firstly, contrasting the DA trend map between Figure 13(e) and Figure 8(e), one can see a prominent pattern change from the Basin DA to Girdded DA, where more content of GRACE TWS observations has been introduced to the Gridded DA results. For example, the positive trend signal (indicating a water gain) found at the heart of MB ($\sim 20°$E, $\sim 45°$N) appears more evident in Figure 13(e) than Figure 8(e), and such a positive signal is quite consistent with that of GRACE-TWS. Likewise, by contrasting Figure 14(e) with Figure 9(e), it is found from the annual amplitude maps that (i) the magnitude has been tuned closer to GRACE-TWS by Gridded DA than Basin DA:





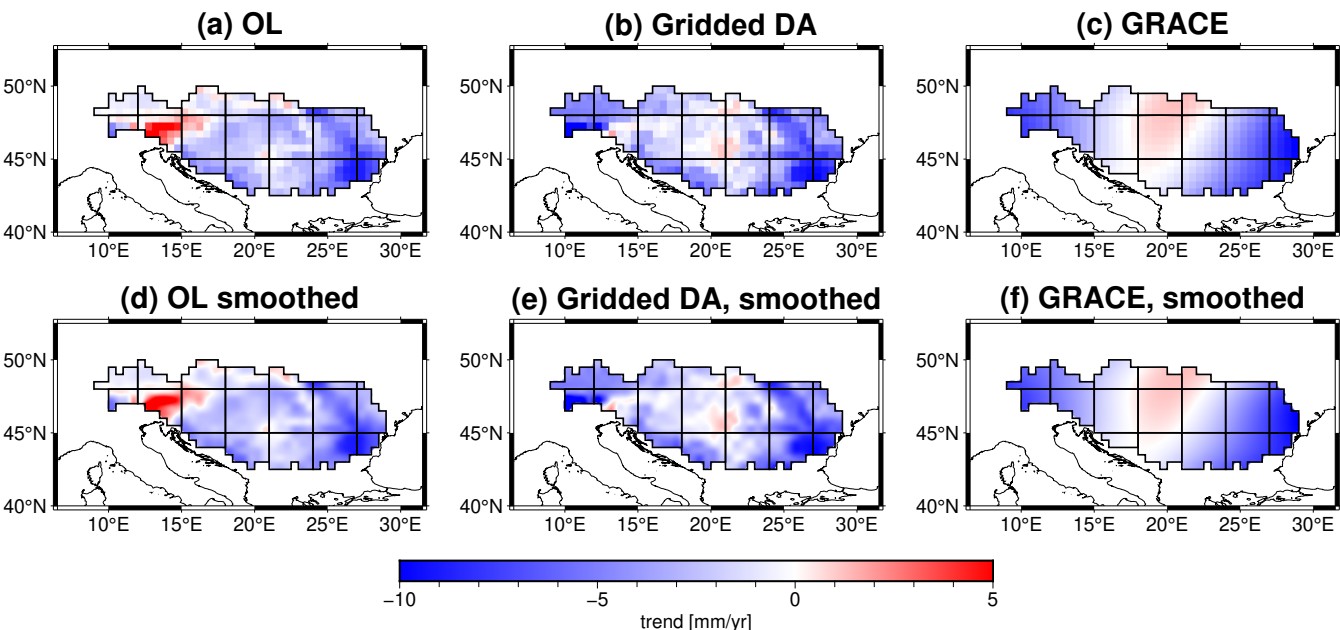

**Figure 13.** As that of Figure 8 but this is for the Gridded DA.

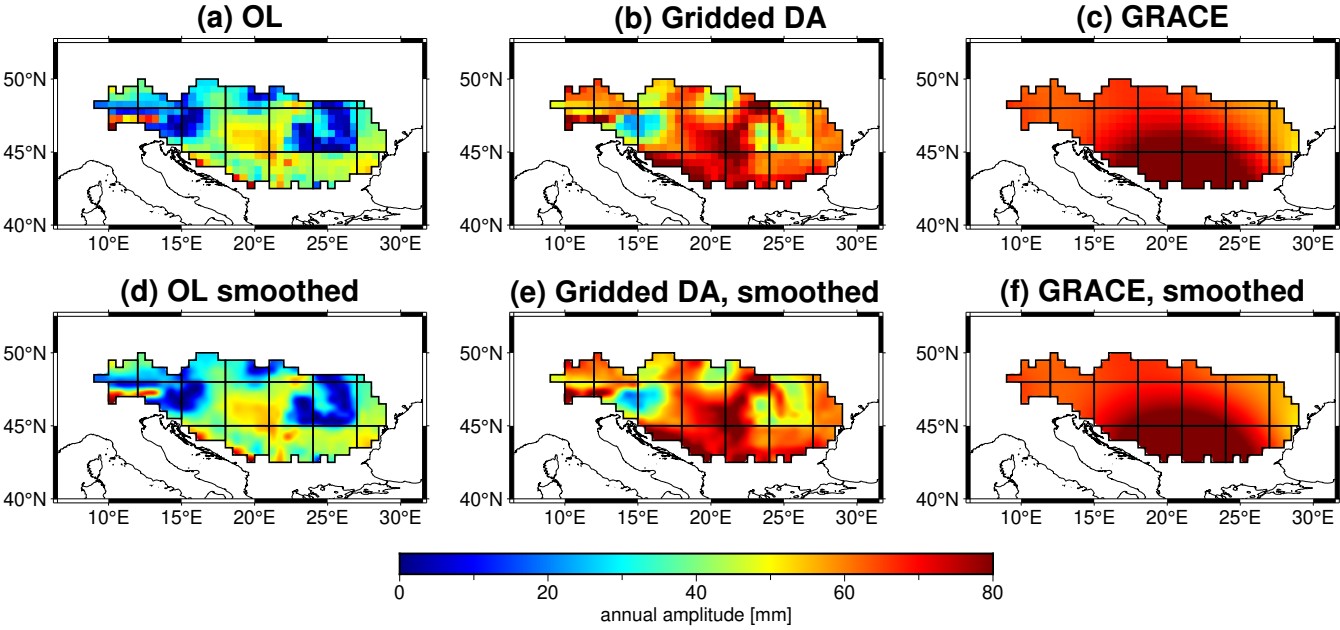

**Figure 14.** As that of Figure 9 but this is for the Gridded DA.

RMS[GRACE minus Gridded DA] is 16.5 mm while RMS[GRACE minus Basin DA] is 22.1 mm; (2) the spatial pattern of

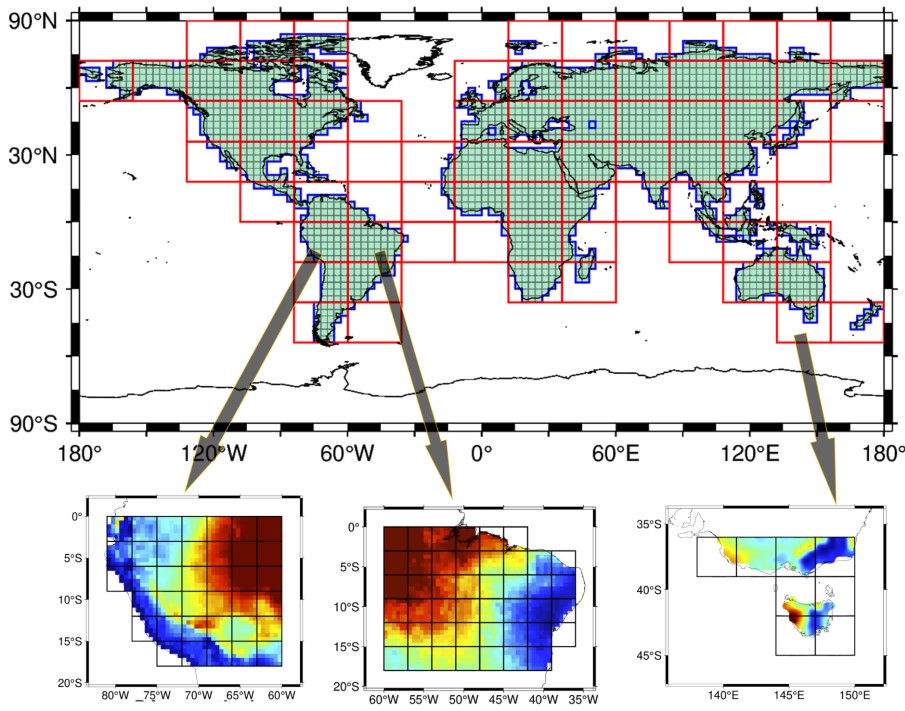

**Figure 15.** A schematic diagram of the GLDA implementation, where extension of the zone (6° as indicated by Figure 4(a)) has been applied but not shown here for simplifying the visualization.

Gridded DA has revealed more deviation with the OL: the spatial correlation (abbreviated as SCOR) of Gridded DA and OL is 0.81, SCOR[Gridded DA versus GRACE] is 0.51, while SCOR[Basin DA versus OL] is 0.88 and SCOR[Basin DA versus GRACE] is 0.45. Based on what we have observed so far, we suppose that by incorporating finer-scale satellite-derived TWS (from basin-scale to grid-scale) into the DA procedure, one would expect to benefit more from TWS observations in terms of both spatial details and the magnitude of the signal. This finding is promising in the context of future satellite gravity mission products, where better spatial and temporal resolution and less complicated error structure are expected (Daras et al., 2023). Nevertheless, this is not meant to assert which solution is better, and a detailed evaluation requires independent validations, which will be addressed in future.

### 4.3   0.1° and daily GLDA

In this case study, we perform the GLDA experiment covering 2002-2010 by merging monthly GRACE TWS into the daily W3RA globally at high spatial resolution, i.e., 0.1° or equivalently 10-12 km, which is likely the highest resolution among similar previous efforts. By this resolution choice, our intention is to test the ability of the proposed PyGLDA platform, whereas the resolution can be indeed flexible. Based on previous experience (Li et al., 2019; Gerdener et al., 2023), we also chose to implement a Gridded DA as has been done in Sec. 4.2. Greenland and Antarctica are excluded because GHMs often





**Table 3.** An overview of the numerical efficiency tests for PyGLDA, where the computation is designed from years 2002-2010 with an ensemble size of 30.

| Step | Sub-step | Target area | Resources needed[1] | Time (hour per core) |
|---|---|---|---|---|
| Data pre-process or preparation | GRACE signal process[2] | globe | 1-core, up to 10 GB | $\sim 0.8$ |
| | GRACE covariance estimation | globe | 1-core, up to 15 GB | $\sim 3.0$ |
| | W3RA data pre-process[3] | 1-patch[4] | 1-core, up to 6 GB | $\sim 0.5$ |
| | Perturbation | 1-patch | 1-core, up to 4 GB | $\sim 0.2$ |
| Data process | Spin-up (2-years) | 1-patch | 1-core, up to 3 GB | $\sim 0.04$ |
| | Open loop | 1-patch | 30 cores, up to 60 GB | $\sim 0.3$ |
| | DA | 1-patch | 30 cores, up to 70 GB | $\sim 0.5$ |
| Data post-process | Statistical analysis | 1-patch | 30 cores, up to 10 GB | $\sim 0.1$ |
| | Global aggregation | globe | 1-core, up to 15 GB | $\sim 1.0$ |
| | Visualization | 1-patch | 1-core, up to 2 GB | $\sim 0.01$ |

[1] CPU cores and running memory for serial/parallel computations.

[2] This includes conversion from GRACE L2 to TWS and necessary corrections as indicated by Tab. 2.

[3] This includes the crop of ERA5-land forcing field (as well as the model parameters and climatology) at intended area, and its upscaling to a desired resolution.

[4] This indicates the performance is evaluated on one patch as defined in Figure 15.

do not cover these regions. The unique advanced configuration of GLDA includes the strategy of global disaggregation and aggregation, which has also been explained in Sec. 3.3. The overall implementation of GLDA is conceptually illustrated in Figure 15, where one can see the global divisions of the patches (in red rectangle), the shape of the 'basin' inside each patch (in blue), and the 'subbasins' inside each 'basin' (in black box). All the shape files related to the aforementioned definitions are shared in the PyGLDA package. In fact, as addressed in Figure 4, each patch has been extended by $6°$, but this is not shown to keep the visualization simple. Then, DA was performed in each patch one by one and added back as indicated by Figure 15, and eventually, all were integrated together as a whole via the proposed weighting method.

Furthermore, the detailed road map of GLDA in our practice is introduced in Tab. 3: how each sub-module of PyGLDA is called to complete a specific task. In general, the flow-work can be simplified as three stages: data pre-processing, data processing and data post-processing, each comprising a few sub-steps. In this way, an evaluation of each module's numerical efficiency becomes easier. Therefore, following the aforementioned configuration, we perform the GLDA tests on our in-house computer cluster, where only one node is utilized, which is equipped with 32 cores (Intel Xeon Gold 6130@1.59 GHz) and 192 GB running memory. While the value (resources consumption) of Tab. 3 is not that accurate (highly depending on the user





case and platform to do the computation), it can roughly represent the PyGLDA's numerical efficiency and address the user's concern on PyGLDA's actual performance.

From Tab. 3, it can be found that the data pre-processing has the largest computation time, followed by the data processing and the data post-processing. Fortunately, while the data pre-processing is computationally expensive, it requires to be run once, and the results can be saved for later use. In addition, all the data pre-processing is based on one single thread, which means that a large reduction of computation time could be achieved through multi-thread parallelization. By contrast, the computation of the data processing, as the core of the GLDA, is super fast and should be affordable even on a personal desktop. In the case

of a regional Gridded DA, such as that for the Danube River Basin in Sec. 4.2, the data processing costs only less than 15 minutes in our test, which should be desirable for most applications. By accounting for the three processes altogether, the time to finish a complete GLDA is roughly 5 days, using only one node of the cluster. One can actually expect a higher efficiency by making use of multiple nodes. Despite the satisfactory efficiency of PyGLDA, implementing a GLDA would generate a considerable amount of temporary data and output (up to 15 TB in this case study), whose temporal resolution is daily scale,

and the vertical storage components consist of six layers (snow, biomass water, surface water, shallow soil water, deep soil water and groundwater) at $0.1°$, globally, i.e., 2142930 grid points.

Then, the TWS results (from the model, GRACE observation and GLDA) are collected in Figure 16. By comparing the trend map between W3RA and GRACE, one can see a very evident underestimation of the TWS trending on a global scale in the model, which indeed was also confirmed by previous studies that current models cannot well reproduce the long-term change

in particularly the groundwater (Scanlon et al., 2018; Mehrnegar et al., 2020; Forootan et al., 2024), and many evidences have shown that the groundwater is rapidly declining worldwide (Jasechko et al., 2024). GRACE TWS shows the trends, but its spatial resolution is too coarse for hydrological interpretation. In short, the model is poor at revealing the magnitude and trends, but it can exploit the spatial resolution. Fortunately, GLDA can use these two in a complementary way, where, for example, in Figure 16, one can see that DA results adopt the resolution of W3RA and the magnitudes are successfully tuned

to GRACE. In statistics, as for the trend map, it is found that the spatial correlation between W3RA and DA is 0.12, which is because the original model does not represent trends well. The correlation between GRACE and DA is increased to 0.35. The global mean RMS (Root Mean Squared) of differences between W3RA and DA is found to be 50.0 mm/yr, whereas that of GRACE and DA is 45.1 mm/yr. As for the annual amplitude, it is found that the spatial correlation between W3RA and DA is 0.86, whereas that between GRACE and DA is 0.92; the global mean RMS of differences between W3RA and DA is 26.0 mm, whereas that between GRACE and DA is 29.0 mm. Based on these results, it seems that the DA relies more on GRACE

than the W3RA model, in terms of both trend and seasonality. Nevertheless, this might not be optimal since the weight of the model also depends on how the ensembles are generated, and proper tuning of the model covariance should be subject to more investigations. In addition, validations of the proposed GLDA against independent data sets are desirable, which will be added in future to PyGLDA.



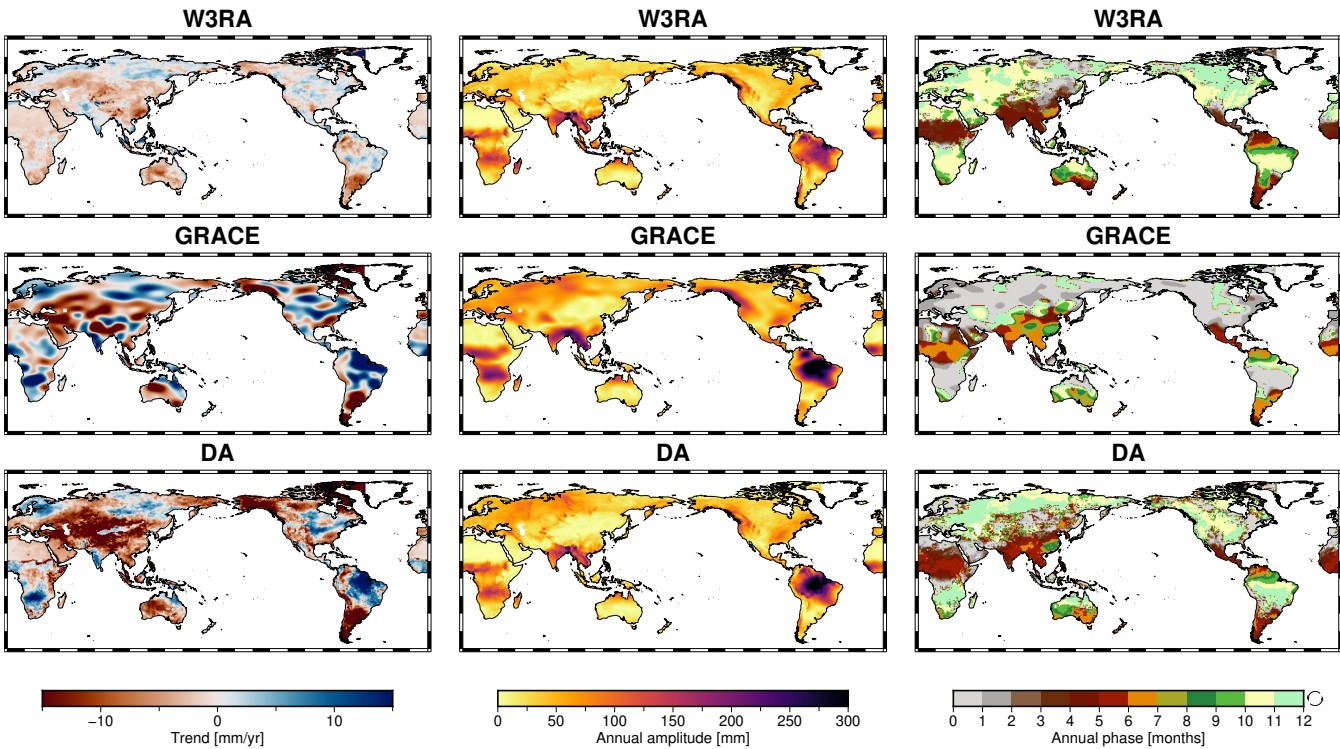

**Figure 16.** TWS results (2002-2010) in terms of secular trend (the 1st column), annual amplitude (the 2nd column) and annual phase (the 3rd column). Plots from the top to the bottom correspond to W3RA (open-loop ensemble-mean model output), GRACE (perturbation-free observations), and the assimilated results. For appropriate visual comparison, the results of W3RA and DA are upscaled from $0.1°$ to $0.5°$ to be consistent with the grid of GRACE TWS used in this study.

## 5    Conclusions

We presented an advanced global and regional land data assimilation system in a Python package called PyGLDA (Yang, 2024). This system is currently used to merge the GRACE TWS observations into Global Hydrological Models (GHMs). Unlike previous publicly available DA frameworks that often focused on the theoretical filter design, our PyGLDA is an open-source system oriented to the full-scale and end-to-end hydrological DA with satellite gravity data such as GRACE, GRACE-FO and those of future gravity missions. Our major contribution is that PyGLDA can be used globally and regionally at various resolutions at arbitrary regions, and in addition, it can also allow users to flexibly switch between grid-scale and basin-scale DA. Various case studies are applied, whether locally or globally, and whether grid-scale or basin-scale, confirming that PyGLDA has fulfilled its goal to acquire reasonable temporal, horizontal, and vertical disaggregation of the coarse TWS estimates of satellite gravity missions at a finer scale. This eventually improves TWS estimates of GHMs. PyGLDA is being developed to allow for more candidate GHMs, other than W3RA that was used in this study, and more observation sources


(multiple satellite remote sensing) to participate in DA to improve possibly some of the TWS compartment (e.g., soil water) instead of the whole.

As a community-level and open-source platform, PyGLDA may hopefully help to contribute to establishing a regional or global land data assimilation for early warning and water management systems. Its flexible architecture (modular structure
to allow for independent treatment/development with the model, the observations and the assimilation), easy communication (modify the setting (.json) file and shape (.shp) file to manage the workflow), advanced programming language (object-oriented Python) make PyGLDA friendly to both users and developers. In particular, global-scale assimilation is challenged with numerical difficulties, and the strategy of PyGLDA may offer an alternative way to address this issue. By furthermore introducing numerical optimization (exploit Numpy broadcast to accelerate computation) and multi-CPU/multi-node parallel computation
via MPI4PY, PyGLDA should be practically helpful to reach the prospective goal of global water monitoring. Nevertheless, there are also some potential caveats of using PyGLDA that need to be fixed soon. (1) Potential numerical instability introduced by the inverse of the covariance matrix has been addressed by previous work (Khaki et al., 2017), but this has not been found yet so far in our study. Therefore, in the current version, none of the covariance localization techniques has been integrated into PyGLDA, and consequently, users may face the potential risk of instability when the gridded DA is configured to
run at a smaller scale, such as $1° \times 1°$. More experiments need to be done to guarantee the robustness of PyGLDA. (2) An oversimplification of the current perturbation strategies in both forcing field and model parameters may lead to a unrealistic uncertainty/covariance of the model, resulting in less optimized weighting between the model and observations. Therefore, accounting for reasonable (e.g., local correlation) spatial and temporal perturbation is under development for PyGLDA.

*Code availability.* PyGLDA is written in Python as an open-source project licensed under MIT, and its latest stable version is publicly
available at Zenodo repository (Yang, 2024), which comes with the code, a guide to installation, and a few demos to showcase various DA experiments. In that repository, we also offer necessary sample data so that one can experience an instant running of all those demonstrations. In addition, PyGLDA is being actively developed under a GitHub repository; see https://github.com/AAUGeodesyGroup/PyGLDA, where one can fetch the newest version. The original Matlab implementation of W3RA water balance model can be found via https://www.dropbox.com/scl/fo/b0hneugr9vao0rqm4oh86/AEPPU-QG6kgh9wTlIBgiwMQ?rlkey=q7ux08mitdghnoac3e4spwaev&e=1&dl=0. Figures of this pa-
per were created through PyGMT using Generic Mapping Tools (GMT) version 6 licensed under BSD 3-clause, available at https://www.genericmapping-tools.org/.

*Data availability.* The shape files (.shp) of DRB and global patches used in this study, as well as other auxiliary data to run PyGLDA demos are available at Yang (2024). GRACE L2 monthly gravity fields are available at ICGEM (International Center for Global Gravity Field Models) via https://icgem.gfz-potsdam.de/home. The ITSG-2018 data (the normal equations in Sinex) used for estimating GRACE-
based TWS's variance-covariance are available at http://ftp.tugraz.at/outgoing/ITSG/GRACE/ITSG-Grace2018/monthly/normals_SINEX/monthly_n96/. The MCMC error propagation follows the methodology of Yang et al. (2024a). The meteorological forcing fields, i.e., ERA5-land, are all downloaded from https://cds.climate.copernicus.eu/cdsapp#!/dataset/reanalysis-era5-land?tab=overview.





## Appendix A: EnKF and EnKS

The Ensemble Kalman Filter (EnKF) is a merger that applies observations to update the model states and parameters based on
their relative errors (Evensen et al., 2009). Errors of the model are generated by a Monte Carlo simulation that considers $i$ (i.e.,
$i = 1, ..., n$) ensemble members of the model parameters ($\Theta_i^f$) expressed as:

$$\mathbf{\Theta}_i = \mathbf{\Theta}_0 + \xi_i, \, i = 1, ... n, \tag{A1}$$

where $\mathbf{\Theta}_{0_{m_1 \times 1}}$ ($m_1$ being the number of parameters) is a vector representing the initial values of the model parameters. Random errors ($\xi_i$) are considered to perturb the initial values of the parameters. The distribution of these errors can be 'Gaussian' and 'Triangle' as (Schumacher, 2016). In the example of PyGLDA as described in Section 4.1, eight model parameters are selected to be perturbed, thus $m_1 = 8$. Perturbations of the forcing fields follow

$$\mathbf{M}_i = \mathbf{M}_0 + r_i, \, i = 1, ... n, \tag{A2}$$

where $\mathbf{M}_{0_{p \times q}}$ ($p$ being the number of grid points, $q$ being the number of forcing field) is a vector representing the initial values of the forcing inputs. Random errors ($r_i$) are considered to perturb the initial values of these errors. In PyGLDA, we consider multiplicative errors for precipitation, thus $q = 1$. The model's forecasting (shown by $^f$ which can be the simulation of W3RA or any other GHMs) ensembles are then generated following

$$\mathbf{X}_i^f = F(\Theta_i, \mathbf{M}_i), \, i = 1, ... n. \tag{A3}$$

For example, we use $n = 30$ ensemble members to perform our experiments of Section 4.

In each step of GLDA (TWS data are introduced), each ensemble of model states (e.g., in forecasting mode) is represented
by $\mathbf{X}_{p \times n}^f$ as:

$$\mathbf{X}^f = (\mathbf{x}_1^f, \cdots, \mathbf{x}_n^f), \tag{A4}$$

where $\mathbf{x}_{i \, p \times 1}^f (i = 1, \cdots, n)$ is the $i$'th ensemble. The ensemble mean is defined as:

$$\bar{\mathbf{x}}^f = \frac{1}{n} \sum_{i=1}^{n} \mathbf{x}_i^f, \tag{A5}$$

and the error covariance matrix of the model forecast ($\mathbf{C}_{m \times m}^f$) is computed as follows:

$$\mathbf{C}^f = \frac{1}{n-1} \sum_{i=1}^{n} (\mathbf{x}_i^f - \bar{\mathbf{x}}^f)(\mathbf{x}_i^f - \bar{\mathbf{x}}^f)^T. \tag{A6}$$

The analysis step (shown by the upper-index $a$) corrects the states and their uncertainties using the GRACE-TWS observations as:

$$\mathbf{X}^a = (\mathbf{x}_1^a, \cdots, \mathbf{x}_n^a) = \mathbf{X}^f + \mathbf{K}(\mathbf{Y} - \mathbf{H}\mathbf{X}^f), \tag{A7}$$

$$\mathbf{C}^a = (\mathbf{I} - \mathbf{K}\mathbf{H})\mathbf{C}^f, \tag{A8}$$





where $Y_{p \times n}$ is an ensemble of actual observations, which are perturbed by the estimated noise from the covariance matrix of observations (Yang et al., 2024a). This means that, for the example of Section 4.2, the dimension is $19 \times 30$, and for the global GLDA (Section 4.3) with $3°$ grid resolution, the dimension is roughly $1700 \times 30$. The analyzed unknowns ($\mathbf{X}^a_{p \times n}$) and their covariance matrix ($\mathbf{C}^a_{p \times p}$) depend on the estimation of the Kalman gain matrix ($\mathbf{K}$) and the design matrix $\mathbf{H}$, which are defined as

$$\mathbf{K} = \mathbf{C}^f \mathbf{H}^T \left( \mathbf{H} \mathbf{C}^f \mathbf{H}^T + \mathbf{C}^r \right)^{-1}, \tag{A9}$$

where $C^r$ is the covariance matrix of the satellite-derived TWS estimates. Nevertheless, the formulation of $\mathbf{H}$ highly depends on the definition of the state vector $\mathbf{X}$. For the W3RA model, in terms of each ensemble member, $\mathbf{X}$ includes seven vertical layers following

$$\mathbf{X} = (\mathbf{s}_1, \cdots, \mathbf{s}_v), \tag{A10}$$

where $\mathbf{s}$ represents one of the surface water, the topsoil water, the shallow soil water, the deep soil water and the groundwater, the biomass water, and the open free water. Nevertheless, as some of these vertical compartments might have more than one column, the dimension ($v$) along vertical profile is 10 in our study. Laterally, the state vector $\mathbf{X}$ adheres to the native spatial resolution of the W3RA model, for example, $0.1°$, and therefore for GLDA in Section 4.3 the dimension is over 2 million at a global scale. In this case, which is exactly the case of EnKF, $\mathbf{H}$ represents the vertical aggregation (into TWS) and the lateral

aggregation (from $0.1°$ to $3°$).

The EnKS we implement is similar to that of Tian et al. (2017), where $\mathbf{X}$ no longer stands for the monthly mean like GRACE-TWS. Instead, $\mathbf{X}$ consists of daily state vector of one month from the model, and in this case, the design matrix $\mathbf{H}$ of EnKS shall be responsible for the temporal aggregation other than the vertical and lateral aggregation of EnKF. The benefit of EnKS is potentially better tuning of daily output by accounting for the temporal covariance, but the price is more computation burden

as the dimension along time increases to 30 (or 28 or 31) from 1 of EnKF. More details on the derivation of the design matrix and EnKS can refer to Schumacher (2016); Tian et al. (2017).

**Appendix B: A summary of input-output of PyGLDA**

Please refer to Table. B1 for a record of software input and output. Details on the inputs (GRACE, GHM, and Mask) can be also found in Section. 2. Information on the setting files (.json) can be found at either our Github or data repository (Yang,

2024). The output can be obtained by running the demo offered in that repository as well.

*Author contributions.* Fan Yang led the writing of the manuscript and the design of the software. Maike Schumacher led the design of DA algorithms, Leire Retegui-Schiettekatte helped with the implementation, Albert van Dijk offered the original W3RA code and helped with technical discussions, and Ehsan Forootan led the project and improved the writing. All authors commented on the manuscript draft.



**Table B1.** A summary of input-output of PyGLDA

| Input | Class | Sub-class |
|---|---|---|
| | GRACE | L1b-L2 normal equations |
| | | L2 monthly gravity fields |
| | | GIA model |
| | | DDK filtering matrix |
| | | Low-degree coefficients |
| | GHM | Forcing field from ERA5-land |
| | | Climatologies |
| | | Parameter fields |
| | Mask (.shp) | GRACE global land mask |
| | | GHM global land mask |
| | | Forcing field global land mask |
| | | GHM running mask |
| | | Basin and subbasin mask |
| | Setting (.json) | Data preprocess |
| | | W3RA-GHM setting |
| | | Perturbation |
| | | Data assimilation |
| **Output** | **Class** | **Sub-class** |
| | GRACE (.h5df) | Grid/Basin-scale TWS |
| | | Perturbed TWS ensemble |
| | | Full variance-covariance |
| | GHM (.h5df) | Initial field (spin-up) |
| | | Cropped input fields |
| | | Perturbed input field ensemble |
| | | Daily state vector ensemble |
| | | Else output, e.g., evaporation |
| | DA (.h5df) | Ensemble mean of state vector |
| | | Ensemble spread of state vector |
| | | Monthly mean output |
| | | Statistics, e.g., basin TWS |

*Competing interests.* The authors have no competing interests to declare that are relevant to the content of this article.

*Acknowledgements.* This work is supported by the Danmarks Frie Forskningsfond [10.46540/2035-00247B] through the DANSk-LSM project. Fan Yang acknowledges financial supports through the national Natural Science Foundation of China (Grant No. 42274112 and No. 41804016).





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
