# Peer review of "PyGLDA: a fine-scale Python-based Global Land Data Assimilation system for integrating satellite gravity data into hydrological models"

_Geoscientific Model Development, 2024_

## Referee Comment (RC2)

[referee-annotated manuscript omitted]

---

## Author Comment (AC1)

**Response to the Reviewer #1 (RC1)**

*General comments:*

*This study presents an open-source global data assimilation (DA) system for integrating GRACE/GRACE-FO total water storage into Global Hydrological Models. The study achieves a previously unseen high spatial resolution, with the potential to increase the use of GRACE/GRACE-FO data in the hydrological modelling community. The study is well within the scope of GMD and a relevant contribution, particularly as it provides a reproducible and transparent approach. The authors have done a thorough effort to test that their implementation can be used at the targeted levels and with consistency.*

**Comment 1:**

*The paper could benefit from a proof-reading and more importantly some clarifications – quite a few terms are treated as self-explanatory which they are not for non-GRACE experts, who might still benefit from using the tools and models developed and kindly made available by the authors.*

**Response**: Thank you for your feedback. We have carefully proofread the manuscript and made several revisions to improve clarity. Specifically, we have provided additional explanations and references for GRACE-related terminology to enhance accessibility for a broader audience. We welcome any further suggestions for improvement.

**Comment 2:**

*While the data processing and DA setups are quite thorough, the GHM used is still quite fundamental to the DA and clearer details on its limitations and structure could be beneficial both to understanding some of the results, but also to identify where additional experiments and developments are warranted. I think the paper would gain in strength and impact if it were to include some model validation and analysis of the hydrological consistency.*

**Response**: We greatly appreciate this constructive suggestion. Our primary goal was to introduce a practical framework for assimilating satellite gravity data into hydrological models, which is why we selected the W3RA model, a relatively simple but well-established GHM used in numerous hydrological DA studies (van Dijk et al., 2014; Tian et al., 2017). These studies include comprehensive validations against soil moisture, river discharge, and in-situ well observations. However, we acknowledge that validating global DA remains a significant challenge due to the lack of reliable external observations at a global scale, see also Li et al (2019).

To address potential limitations about the employed GHM, i.e., W3RA. Here, we highlight the following points:

1. Lack of Anthropogenic Water Use: The model does not account for anthropogenic influences such as irrigation, which significantly impact the global water cycle at somewhere. For instance, in regions like the North China Plain, groundwater over-extraction for irrigation is a major cause of water decline, which is not well represented in W3RA. Consequently, the model is less reliable at these regions. However, the model covariance generated from ensemble cannot

well reflect the model deflection at these implausible places. This issue may lead to a higher weight of model in DA, resulting in an underestimation of the long-term water variability over those places.

2. Inability to Simulate Water Changes in Reservoirs and Lakes: Although our DA method excludes the lakes, the spatial leakage of satellite gravity data can introduce artificial signals into adjacent areas, affecting DA results in regions with significant lake water changes, such as the areas neighboring Caspian Sea.

3. Glacier Representation Issues: Despite having a snow module, W3RA does not adequately capture water storage changes in glacier regions. For example, the strong ice-melting signal over Alaska is detected by GRACE but is not well represented in W3RA, which lacks a glacier module. This may lead to inaccurate vertical disaggregation of water storage at glacier regions.

These limitations may degrade DA results, necessitating future improvements. Possible solutions include:

1. Replacing W3RA with a more advanced hydrological model, such as WaterGAP 2.2e (Müller et al., 2024), which accounts for anthropogenic water use, reservoirs, lakes, and glaciers. This model is released in Python and can be compatible with our PyGLDA framework.

2. Enhancing the DA strategy by applying an empirical inflation factor (Schumacher, 2017) to amplify the W3RA covariance at locations known to be less reliable (e.g., glacier regions and areas with intensive anthropogenic water use)

3. Mitigating lake-induced leakage issues by incorporating altimetry data to model and remove lake water changes from GRACE observations (Deggim et al, 2021).

We have integrated these discussions into a newly added "Limitations" section in the revised manuscript.

References:
- Tian, S., Tregoning, P., Renzullo, L. J., van Dijk, A. I. J. M., Walker, J. P., Pauwels, V. R. N., and Allgeyer, S.: Improved water balance component estimates through joint assimilation of GRACE water storage and SMOS soil moisture retrievals, Water Resources Research 53, 1820–1840, https://doi.org/10.1002/2016WR019641, 2017.
- van Dijk, A. I. J. M., Renzullo, L. J., Wada, Y., and Tregoning, P.: A global water cycle reanalysis (2003-2012) merging satellite gravimetry and altimetry observations with a hydrological multi-model ensemble, Hydrology and Earth System Sciences, 18, 2955–2973, https://doi.org/10.5194/hess-18-2955-2014, 2014.
- Li, B., Rodell, M., Kumar, S., Beaudoing, H. K., Getirana, A., Zaitchik, B. F., de Goncalves, L. G., Cossetin, C., Bhanja, S., Mukherjee, A., et al. (2019). Global GRACE data assimilation for groundwater and drought monitoring: Advances and challenges, Water Resources Research, 55, 7564–7586, https://doi.org/10.1029/2018WR024618
- Müller Schmied, H., Trautmann, T., Ackermann, S., Cáceres, D., Flörke, M., Gerdener, H., ... & Döll, P. (2024). The global water resources and use model WaterGAP v2. 2e: description and evaluation of modifications and new features. Geoscientific Model Development, 17(23), 8817-8852.
- Schumacher, M. (2016). Methods for assimilating remotely-sensed water storage changes into hydrological models, Ph.D. thesis, Universitäts-und Landesbibliothek Bonn.
- Deggim, S., Eicker, A., Schawohl, L., Gerdener, H., Schulze, K., Engels, O., Kusche, J., Saraswati, A. T., van

Dam, T., Ellenbeck, L., Dettmering, D., Schwatke, C., Mayr, S., Klein, I., and Longuevergne, L. (2021). RECOG RL01: correcting GRACE total water storage estimates for global lakes/reservoirs and earthquakes, Earth System Science Data, 13, 2227–2244, https://doi.org/10.5194/essd-13-2227-2021.

*Specific comments:*

**Comment 3:**

*l. 28-30 I would revise how you classify the models – most of these are rainfall-runoff and conceptual which is also the case of a water balance model. In general we talk more of conceptual versus physically based and whether a model is lumped, distributed or semi-distributed (e.g. SWAT with Hydrological Response Units). The question can also be how many processes are lumped/conceptualized and how simple the model structure is.*

**Response**: Thank you for this insightful suggestion. We have revised the introduction of global hydrological models in accordance with standard classifications as suggested.

**Comment 4:**

*2.2 Which up- or down-scaling routine are you using?*
*2.3 How are the parameters down-scaled? Is there any risk of non-linearities at the boundaries between pixels or basins?*
*Would refining the grid size not increase your susceptibility to lateral redistribution? Does W3RA include any kind of routing component?*

**Response**: We use ECMWF's official 'metview' package for up/down-scaling, which ensures physical consistency and minimizes boundary effects. This is a free toolbox and can be obtained from https://metview.readthedocs.io/en/latest/index.html. In addition, ECMWF provides a technical document that details its interpolation (up/down-scaling) strategy, including mass conservation and boundary treatment to maintain physical consistency (https://confluence.ecmwf.int/display/OPTR/Introduction+to+ECMWF+computing+services+and+MARS+2017?preview=%2F73008494%2F73014654%2Fintro_interpolation_2017.pdf).

While no up/down-scaling method is perfect, we believe 'metview' provides a robust solution. Additionally, refining grid size may increase susceptibility to lateral redistribution, but this does not impact our study since GRACE DA primarily focuses on vertical storage changes. W3RA does include a river routing module, but it is not used here as GRACE measures total water storage without distinguishing lateral redistribution components. The river routing module of W3RA is available at https://www.dropbox.com/scl/fo/b0hneugr9vao0rqm4oh86/ABdQ9vV7-FGRyb7STgw-7ew/W3_code/W3_Princeton?dl=0&rlkey=q7ux08mitdghnoac3e4spwaev&subfolder_nav_tracking=1.

**Comment 5:**

*2.4 Please explain more clearly what ITSG-2018 is and who is developing it? In 2.4 you only present the three official data centers, and it would be good with a bit more background information on ITSG-2018 and why you prefer an unofficial provider (and whether that is available to any user). Why is a better quality expected with ITSG-2018?*

**Response**: ITSG-2018 is developed by Prof. Mayer-Gürr's team at Graz University, known for their outstanding work at developing advanced methods in gravity field recovery from GRACE

(https://www.tugraz.at/institute/ifg/downloads/gravity-field-models/itsg-grace2018/).

The GRACE community widely recognizes ITSG-2018 for achieving the highest signal-to-noise ratio among existing GRACE-derived gravity field products, please refer to the evaluation paper (Meyer et al., 2019) for more details. Its superior quality results from a more refined computation method that accounts for additional uncertainty sources (Kvas et al., 2019). Furthermore, official data producers plan to adopt similar improvements in their next updates, as discussed at the GRACE Science Team Meeting 2024 (https://www.gstm-2024.eu/). ITSG-2018 is publicly available via FTP, and we have now included access details in the "Data Availability" section of the revised manuscript (https://dataservices.gfz-potsdam.de/icgem/showshort.php?id=escidoc:3600910)

References:

- Meyer, U., Jean, Y., Kvas, A., Dahle, C., Lemoine, J.M., Jäggi, A., 2019. Combination of GRACE monthly gravity fields on the normal equation level. Journal of geodesy 93, 1645–1658. doi:10.1007/s00190-019-01274-6
- Kvas, A. and Mayer-Gürr, T., 2019. GRACE gravity field recovery with background model uncertainties, Journal of Geodesy, 93, 2543–2552, https://doi.org/10.1007/s00190-019-01314-1

**Comment 6:**

*Table 1 – please give more details on the corrections which use acronyms or simple symbols, as well as what each correction does.*

**Response**: Thanks for the suggestion. We have included brief descriptions of each correction in the revised manuscript and referenced a previous publication (Liu et al., 2025) for further details.

- Liu, S., Yang, F., and Forootan, E.: SAGEA: A toolbox for comprehensive error assessment of GRACE and GRACE-FO based mass changes, Computers Geosciences, 196, 105 825, https://doi.org/10.1016/j.cageo.2024.105825, 2025.

**Comment 7:**

*l. 206 Please clarify – do you mean that if the required number of ensembles is run, each individual ensemble is independent of each other or is that the case no matter the number of ensembles?*

*l. 214 What would be a recommended number of ensembles?*

**Response**: Sorry for our unclear statement. PyGLDA executes ensemble runs in parallel, ensuring that each ensemble member operates independently, regardless of the number of ensemble members. Based on previous studies (Schumacher, 2016), an ensemble size of 30 members is recommended as a practical balance between accuracy and computational efficiency. We have clarified this in the revised manuscript.

References:

- Schumacher, M. (2016). Methods for assimilating remotely-sensed water storage changes into hydrological models, Ph.D. thesis, Universitäts-und Landesbibliothek Bonn.

**Comment 8**:

*Figure 3 You are assuming no inter-correlation within the tiles but there is clearly hydrological correlation between several neighboring tiles crossing large river basins – can you clarify how that is not an issue?*

**Response**: We totally agree with you that there must be correlation between the adjacent tiles, which has been actually acknowledged as well in our paper. This issue is also our motivation to develop a new strategy in Section 3.3.2 of the manuscript, by introducing a 'transition zone' to account for the correlation and smooth boundary discontinuities.

**Comment 9**:

*3.3.2 How do you determine the width of the transition zone? Are there large differences where the average might not make sense? Have you tested whether your assumption of reasonability of the inverse-distance weight is correct?*

**Response**: Thanks very much for sharing the concern. We have clarified this methodology and justified our choice of weighting schemes in the revised manuscript:

1. The width of the transition zone is actually flexible and can be changed to any desired value. However, in the paper, we suggest to set the width as 6 degrees, based on empirical correlation length analysis from our previous GRACE studies (Yang et al., 2024).

2. We illustrate the correlation coefficient at a few sample points, see Figure **1**. Given that the region within the central peak (in red) is area of significant correlation, one can find from Figure **2**(a-c) that the meaningful correlation length (half of the range of the peak) is between 9/2 (Figure **3**c) and 14/2 degrees (Figure **4**a), along the east-west profile. Furthermore, the correlation length is 9/2 degrees along the north-south profile. Therefore, from a perspective of 2-D gridded map, it should be safe enough to set an extended area of 6 degree as the 'transition zone'.

3. To merge the adjacent tiles together, we suggest a weighted average. This makes sense when the correlation decays with the distance linearly, and this is actually an reasonable assumption that has been utilized by many works on covariance modeling (Olivier et al., 2022). In addition, in our practice, the differences of two adjacent tiles at the transition zone are actually small because of the constraints of same model and observations at transition zones. Therefore, it is safe to perform an average.

4. This assumption of inverse-distance weight can also be supported by Figure **5**, where the correlation approximately decays linearly within the area of central peak (in red), indicating an inverse-distance weight. Nevertheless, it should be more reasonable to assume a decay in Gaussian manner, which is a widely accepted assumption for covariance localization. For example, a typical covariance localization is the Schur product (Webber & Morzfeld, 2023), whose basic idea is to damp long range correlations based on the assumption that correlations decay with distance as Gaussian weighting. In our next update, we shall consider to use Gaussian weighting as the merger of adjacent tiles. Thanks very much for your inspiration.

[Figure]

**Figure 6**: GRACE's correlation coefficient as a function of the distance between the observation point and the center. The center is denoted as the red star in each subfigure, and (a-c) indicates the correlation along the east-west profile, while (d) indicates the correlation along the north-west profile.

References:
- Yang, F., Forootan, E., Liu, S., & Schumacher, M. (2024). A Monte Carlo propagation of the full variance-covariance of GRACE-like level-2 data with applications in hydrological data assimilation and sea-level budget studies. Water Resources Research, 60, e2023WR036764. https://doi.org/10.1029/2023WR036764
- Olivier Ledoit. Michael Wolf. (2022). Quadratic shrinkage for large covariance matrices. Bernoulli 28 (3) 1519 - 1547, https://doi.org/10.3150/20-BEJ1315.
- Webber, R. J., & Morzfeld, M. (2023). Localized covariance estimation: A Bayesian perspective. arXiv preprint arXiv:2301.04828

**Comment 10:**

*l. 334 Please clarify "the phase shift has been compromised" – where is it happening in the model?*

**Response**: We supplemented an experiment to calculate the phase of model, GRACE observation and DA results over Danube River Basin (DRB). To be specific, the seasonality, i.e., annual variability, is investigated. The statistics is collected in Table 1, from which one can see that the DA has a phase between model and GRACE. This confirms that the DA is tuning the model phase in term of the seasonality. The statistical evidences are added to the revised manuscript as well.

**Table 1**: Annual phase (degrees, range from -180 to 180) of model, observation and DA

| Basin/sub-basin | Model | GRACE | DA |
|---|---|---|---|
| DRB | -42 | 29 | 11 |
| DRB-UB | -85 | 31 | -15 |
| DRB-MB | -42 | 30 | 15 |
| DRB-LB | -37 | 25 | 11 |

**Comment 11:**

*l. 377 Is the water balance also maintained?*

**Response**: The water balance is not maintained any longer after DA, which is indeed a known and mutual issue for nearly all the existing DA studies. Some of previous efforts considers to develop particular techniques for a compensation, for example, using an uncertain constraint (Khaki et al., 2017). However, such technique is unique and has not been adopted in our system yet.

Reference:

- Khaki M, Ait-El-Fquih B, Hoteit I, Forootan E, Awange J, Kuhn M. A two-update ensemble Kalman filter for land hydrological data assimilation with an uncertain constraint. Journal of Hydrology. 2017 Dec 1;555:447-62. https://doi.org/10.1016/j.jhydrol.2017.10.032

**Comment 12:**

*l. 439 Have you considered using GRDC discharge data for validation?*

**Response**: Thanks for the suggestion. In fact, we are planning to develop a new high-resolution assimilated dataset in next paper, where we shall combine the river discharge, soil moisture and GNSS data together for a comprehensive validation. In this paper, we feel that it might be appropriate to focus more on the technical issues like data processing, DA setup and software development, so we have not performed a validation yet. Nevertheless, one can also refer to previous works by our co-authors (van Dijk et al., 2014; Tian et al., 2017), who also assimilated satellite gravity into the W3RA model on local/global scales at lower resolution with similar methods like ours. In those works, one can find many solid evidences including discharge data to validate the effectiveness of DA system:

- Tian, S., Tregoning, P., Renzullo, L. J., van Dijk, A. I. J. M., Walker, J. P., Pauwels, V. R. N., and Allgeyer, S.: Improved water balance component estimates through joint assimilation of GRACE water storage and SMOS soil moisture retrievals, Water Resources Research 53, 1820–1840, https://doi.org/10.1002/2016WR019641, 2017.
- van Dijk, A. I. J. M., Renzullo, L. J., Wada, Y., and Tregoning, P.: A global water cycle reanalysis (2003-2012) merging satellite gravimetry and altimetry observations with a hydrological multi-model ensemble, Hydrology and Earth System Sciences, 18, 2955–2973, https://doi.org/10.5194/hess-18-2955-2014, 2014.

**Comments related to the language and visualization:**

*l. 71 spelling of project name*

*Figure 1. Several of the subparts are difficult to read even when zooming in on a screen (figure in b, boxes in the ensemble output) – I recommend that you either remove the text or increase the font size.*

*l.186 spelling of Gaussian*

*l. 189 verb missing in sentence*

*l 325 This we would expect – the GW has much longer memory than the surface components.*

**Response**: We have implemented all suggested language improvements and enhanced figure readability. Thank you once again for your valuable comments. We hope our revisions address your concerns and improve the clarity and impact of our manuscript.

**Response to the Reviewer #2 (RC2)**

*The manuscripts describes a Python-based Global Land Data Assimilation (PyGLDA) software for integrating the estimates of Terrestrial Water Storage (TWS) variations from satellite gravimetry data into a Global Hydrological Model (GHM). The former data are provided by GRACE and GRACE Follow-on (GFO) satellite missions. As far as the hydrological model is concerned, the authors have adapted W3RA model (line 84). Technically, the Ensemble Kalman filter (EnKF) and an Ensemble Kalman Smoother (EnKS) can be used as data assimilation tools (line 90). In general, the manuscript provides a sufficiently detailed description of the developed software. Nevertheless, there are a number of points which remain unclear. In particular, some statements in the abstract seem to be inconsistent with the main text. This makes an impression that the actual status of the developed software is much less advanced than the abstract claims. I would like to highlight the following uncertain points:*

**Comment 1:**
*1. In the abstract, the authors mention that "choice is made ...to avoid numerical problems, e.g., instabilities related to the inversion of covariance matrices" (lines 5-6). At the same time, they admit at the end of the manuscript that "Potential numerical instability introduced by the inverse of the covariance matrix ...has not been found yet so far in our study. Therefore, in the current version, none of the covariance localization techniques has been integrated into PyGLDA, and consequently, users may face the potential risk of instability when the gridded DA is configured to run at a smaller scale" (lines 466-470). These two statements seem to contradict to each other.*

**Response**: Thanks for pointing out the potential inconsistency. We have clarified that while covariance matrix inversion instability is a known challenge in data assimilation (Gerdener et al., 2023), we have not encountered it in our current study due to specific strategies we employed, such as:

1. Performing DA at a basin scale or at a relatively coarse resolution (3-degree grid), both of which reduce the risk of instability.
2. Using a patch-based approach to divide the global DA domain into smaller regions, further mitigating instability risks.

These methods have been confirmed effective to reduce the illness of covariance matrix (Schumacher et al., 2016; Khaki et al., 2017) and that's why we have not encountered any instability issue now.

However, be aware that the methods are different from traditional 'covariance localization', since which often manipulates the covariance matrix directly by removing spurious correlations based on distance. For example, a typical covariance localization is the Schur product (Webber & Morzfeld, 2023), whose basic idea is to damp long-range correlations based on the assumption that correlations decay with distance as Gaussian weighting. And we think it is still necessary to consider developing such 'covariance localization' in future update of PyGLDA, because:

1. One would like to perform DA at a smaller grid, like 1 degree, and the illness of covariance might increase significantly.

2. Future gravity mission enables a higher resolution application, like at 1 degree.

Thanks very much for your careful review, and we have added explanations at the end of the paper to avoid potential confusion for readers.

References:

- Gerdener, H., Kusche, J., Schulze, K., Döll, P., and Klos, A.: The global land water storage data set release 2 (GLWS2. 0) derived via assimilating GRACE and GRACE-FO data into a global hydrological model, Journal of Geodesy, 97, 73, https://doi.org/10.1007/s00190 023-01763-9, 2023.
- Schumacher, M., Kusche, J., and Döll, P.: A systematic impact assessment of GRACE error correlation on data assimilation in hydrological models, Journal of Geodesy, 90, 537–559, https://doi.org/10.1007/s00190-016-0892-y, 2016.
- Khaki, M., Schumacher, M., Forootan, E., Kuhn, M., Awange, J. L., and van Dijk, A. I.: Accounting for spatial correlation errors in the assimilation of GRACE into hydrological models through localization, Advances in Water Resources, 108, 99–112, https://doi.org/10.1016/j.advwatres.2017.07.024, 2017.
- Webber, R. J., & Morzfeld, M. (2023). Localized covariance estimation: A Bayesian perspective. arXiv preprint arXiv:2301.04828

**Comment 2**:

*2. The abstract claims that W3RA hydrological model is used just to demonstrate the performance of the developed data assimilation software (lines 15-16). At the same time, the manuscript itself does not go beyond W3RA. Moreover, the concluding section mentions that "PyGLDA is being developed to allow for more candidate GHMs, other than W3RA" (lines 454-455). Thus, it remain unclear to what extent the developed software in the present form is suited to assimilate GRACE/GFO data into other hydrological models.*

**Response**: We appreciate this observation. Our clarification is as follows:

1. The goal of PyGLDA is to establish a user-friendly and flexible framework for DA in hydro-geodesy. Therefore, it has been designed in modular structure at a loose coupling that each major module can run independently. For instance, the module of 'observation' can be performed independently to guarantee our next update that shall include multi-sensors, i.e., soil moisture and GNSS. And in our latest update, we have integrated GRACE-mascon (Save et al., 2016) already.

2. Likewise, GHM is also a module that can run independently. At our Github repository (https://github.com/AAUGeodesyGroup/PyGLDA/blob/master/demo/demo_2.py), we have a demonstration (demo-2) to show how to run the model (W3RA) independently. Therefore, changing another GHM should be straightforward for PyGLDA, but only in theory.

3. In fact, we have some practice of migrating other GHM into our DA platform and it works. This GHM is called W3v21 (Earth2Observe, 2017, Section 4.7), which is an advanced version of W3RA but has completely different nature and configuration. However, while W3v21 is an effective tool for simulating soil moisture, this model is found to have severe defects at groundwater simulation at somewhere. Therefore, W3v21 is removed from PyGLDA temporarily at this version.

4. Tests using other Python-based GHMs, like the latest WaterGap model (Müller et al., 2024),

are also being under investigation now, which may produce an updated version of PyGLDA in future.

To avoid overstatement, we have revised the abstract and conclusion to specify that while PyGLDA is designed for multi-GHM compatibility, this study primarily focuses on W3RA.

References:

- Earth2Observe (2017). Report on the improved Water Resources Reanalysis, https://doi.org/10.13140/RG.2.2.14523.67369
- Müller Schmied, H., Trautmann, T., Ackermann, S., Cáceres, D., Flörke, M., Gerdener, H., ... & Döll, P. (2024). The global water resources and use model WaterGAP v2. 2e: description and evaluation of modifications and new features. Geoscientific Model Development, 17(23), 8817-8852.
- Save, H., Bettadpur, S., & Tapley, B. D. (2016). High-resolution CSR GRACE RL05 mascons. Journal of Geophysical Research: Solid Earth, 121 (10), 7547-7569. doi:10.1002/2016JB013007

**Comment 3:**

*3. The authors mention that the adopted hydrological model W3RA ignores lateral water redistribution between grid cells (lines 174-175). This sounds odd. I cannot imagine a hydrological model that ignores streams and rivers. Perhaps, the authors mean the absence of underground water redistribution?*

**Response**: You are correct. W3RA includes a river routing module, which can be run in real-time or as a post-processor. However, since our study focuses on the DA of vertical water storage, we did not activate this module. We have revised the manuscript for clarity.

**Comment 4:**

*4. In the description of data assimilation procedure (Sect. 3.2 and further; also, Fig. 2), the authors seem to use the term "ensemble" instead of the term "ensemble member". This makes the description not very clear.*

**Response**: Thanks very much for your careful review, and we are sorry for our inaccurate use of the terminology. We clarify that, in most situations, we intend to say 'ensemble member' instead of 'ensemble', and there is indeed only one ensemble throughout the DA. We have carefully revised the manuscript to use the correct terminology, ensuring that "ensemble member" is used where appropriate.

**Comment 5:**

*5. At many places in the main text, the authors address correlations (or covariances), but do not explain whether they mean signal correlations or error correlations.*

**Response**: We acknowledge this ambiguity. In GRACE data, the covariance matrix represents a combination of signal and error due to filtering processes. Here are the details:

1. In general, nearly all the existing standard level-2 GRACE temporal gravity fields are obtained from an ordinary least-square (OLS) solution, see our previous work on temporal gravity field modeling from GRACE (Yang et al., 2017).
2. According to the theorem of OLS, the covariance of estimated parameter is propagated from observation residuals, see the OLS document in provided link at below. As a result, the

obtained covariance of GRACE temporal gravity field can be attributed as 'error' covariance. Please also be aware that, OLS assumes the signal (parameters) to be non-stochastic, so that there is actually no concept of 'signal' covariance at all in this case, see the OLS document. As an evidence, we illustrate the standard deviation (square root of the diagonal entries) of GRACE spatial covariance matrix in Figure 7a, and one can clearly see that such a covariance solely reflects the error, which is up to over 100 mm that is much larger than the known hydrological signals.

3. Since gravity field obtained from GRACE contains considerable noises, it is often recommended to apply a filtering to damp the noise. And the nature of the filtering process is actually equivalent to be implementing a regularization, see Kusche et al(2007) and our previous work on GRACE's filtering design (Yang et al., 2024a).

4. The regularization, as a Bayesian process (so that the signal is stochastic), introduces the signal covariance as prior information to stabilize the solution and consequently damp the noise. For example, the DDK filter is a typical regularization filtering, which is also adopted in our study to de-noise GRACE data. The signal covariance of DDK is approximated from long-term hydrological model outputs, see Kusche et al (2007).

5. Due to the filtering, a propagation of GRACE's covariance matrix must be done. This has been realized by a Monte-Carlo method developed in our previous work (Yang et al., 2024b). Since the nature of filtering is a Bayesian process, the obtained covariance after filtering is not indicating solely the 'error' any longer. Instead, it is a combination of the prior information (signal) and noise (error) under an optimal compromise because of the filtering. Please refer to Figure 8b, where one can see that the standard deviation (i.e., the square root of diagonal entries of the covariance matrix) has a close spatial pattern as the signal somewhere, such as Ganges and Amazon. This is because at these places the signal covariance surpasses the error covariance, so that the signal covariance dominates. Such phenomenon has been confirmed by many work as well, demonstrating that filtering introduces an inevitable bias because of introducing signal covariance, see Klees et al (2008).

Due to these reasons, the covariance matrix mentioned in the original manuscript should be indicating a covariance of both signal and error due to the applied filtering. And such covariance can comprehensively reflect the ability of GRACE to monitor TWS anomaly. Therefore, it is likely inaccurate to claim the covariance as either signal or error. In addition, in general DA (or Kalman filtering) system, it is more often to use the term 'model covariance' and 'observation covariance'. GRACE data, as a combination of both signal and error, is sent to DA as the observation. Therefore, in this regard, it might be more accurate to call GRACE's covariance as the observation covariance. Thanks very much for reviewer's suggestion, and we have made the corresponding correction throughout the paper for this concept.

[Figure]

**(a) Before filtering**

**(b) After DDK3 filtering**

**Figure 9**: GRACE spatial covariances (the standard deviation is demonstrated) before and after filtering.

References:

- Yang, F., J. Kusche, E. Forootan, and R. Rietbroek. (2017). Passive-ocean radial basis function approach to improve temporal gravity recovery from GRACE observations, J. Geophys. Res. Solid Earth, 122, 6875–6892, doi:10.1002/2016JB013633

- Yang, F., Liu, S. & Forootan, E. (2024a). A spatial-varying non-isotropic Gaussian-based convolution filter for smoothing GRACE-like temporal gravity fields. Journal of Geodesy 98, 66. https://doi.org/10.1007/s00190-024-01875-w

- Yang, F., Forootan, E., Liu, S., & Schumacher, M. (2024b). A Monte Carlo propagation of the full variance-covariance of GRACE-like level-2 data with applications in hydrological data assimilation and sea-level budget studies. Water Resources Research, 60, e2023WR036764. https://doi.org/10.1029/2023WR036764

- OLS. https://web.stanford.edu/~mrosenfe/soc_meth_proj3/matrix_OLS_NYU_notes.pdf

- Kusche J (2007) Approximate decorrelation and non-isotropic smoothing of time-variable GRACE-type gravity field models. J Geodesy 81:733–749. https://doi.org/10.1007/s00190-007-0143-3

- Klees R (2008) The design of an optimal filter for monthly GRACE gravity models. Geophys J Int 175(2):417–432. https://doi.org/10.1111/j.1365-246X.2008.03922.x

**Comment 6:**

*6. The authors mention that they use a multiplicative error to disturb, among other, forcing fields (e.g., lines 312-314). Furthermore, they mention that the forcing fields include the daily precipitation, the maximal/minimal temperature and the surface solar radiation (lines 123-124). On the other hand, appendix A admits that the only disturbed forcing field is precipitation: "In PyGLDA, we consider multiplicative errors for precipitation, thus q = 1" (lines 499-500). This is inconsistent.*

**Response**: We apologize for this inconsistency. In our study, we perturbed multiple forcing variables (i.e., precipitation, temperature and short-wave solar radiation) and several model parameters. We have revised Appendix A to reflect this and clarify our approach.

In fact, PyGLDA can perturb both the forcing field and model parameters flexibly. In addition, the method of generating perturbation is also optional between Gaussian or Triangular distribution, additive or multiplicative manner. One can refer to our Github repository for the setting file (called perturbation.json) that lists all perturbation options (https://github.com/AAUGeodesyGroup/PyGLDA/tree/master/settings/demo_3).

Since there is no agreement on the 'optimal' perturbation, we do not fix the strategy, and this may lead to inconsistent description. And in the appendix A, we intend to introduce the fundamentals of EnKF, where the precipitation is taken as example, but it does not mean that we only perturbed precipitation. We are sorry for this confusion and have made a correction accordingly.

**Comment 7**:
*7. The statement cited in the previous item implies that no disturbances are applied when the nominal precipitation is zero and no disturbances are applied at all to the other forcing fields. This seems to be sub-optimal, since it may result in an insufficient variability of forcing fields (and, therefore, of the state vectors) among the ensemble members.*

**Response**: Thanks for pointing our the potential risk of our DA configuration. We carefully designed our perturbation strategy to ensure adequate ensemble spread. In addition to perturbing forcing field (precipitation, temperature and short-wave solar radiation), we also perturb nine key model parameters, see Table **2** at below. This was not detailed in the manuscript since this is simply a case study and our choice is also empirical. It is not mandatory that users have to follow our configuration; instead, users can flexibly adjust the perturbation to their specific needs.

**Table 3**: A list of W3RA model parameter to be perturbed for our case studies

| Variable | Generate Perturbation | Error distribution | Perturbation limit |
| --- | --- | --- | --- |
| ER_frac_ref | Multiplicative | Triangle distribution | 30% |
| FsoilEmax | Multiplicative | Triangle distribution | 30% |
| InitialLoss | Multiplicative | Triangle distribution | 30% |
| PrefR | Multiplicative | Triangle distribution | 30% |
| S0FC | Multiplicative | Triangle distribution | 30% |
| S_sls | Multiplicative | Triangle distribution | 30% |
| SdFC | Multiplicative | Triangle distribution | 30% |
| SsFC | Multiplicative | Triangle distribution | 30% |
| Beta | Multiplicative | Triangle distribution | 30% |

Furthermore, in our practices, we can obtain an ensemble with a satisfied spread based on given configuration. Here is an example of our perturbed TWS in Figure 10, where one can find out that the spread among the ensemble members is considerable. Therefore, we think our perturbation strategy is safe for DA.

[Figure]

**Figure 11**: An ensemble (in shaded curves) of TWS by perturbating forcing field and parameters. The blue curve denotes the unperturbed TWS.

**Comment 8:**

*In addition, the manuscript suffers from numerous textual deficiencies. Among others, there are many very long sentences, which definitely should be split into 2-3 shorter ones. Occasionally, the manuscript makes an impression that the authors have never taken time to read the text they wrote. In the enclosed annotated manuscript, I left a number of specific suggestions concerning an improvement of the text. Nevertheless, I encourage the authors also to carefully revise the manuscript themselves.*

**Response**:

We appreciate the reviewer's detailed proofreading. We have performed extensive revisions to improve sentence structure, clarity, and readability. We are looking forward to any further feedback, and we are very pleased to make a further correction.

In addition, the annotations you left in the manuscript has been addressed as well, please see our updated version of the manuscript. Here in this letter, we briefly address some of your annotations that we suppose are relevant.

*The figure (Fig. 4) will look better if panel (c) is shown to the right from panel (a), not below it. In this way, it will be possible to align the horizontal boundaries between patches in panels (a) and (c). Currently, such an alignment is visible for vertical boundaries (panels a and b), but not for horizontal boundaries.*

Corrected as suggested.

*Correlated how? More detail would be appreciated.*

The spatial correlation of perturbation is assumed to be 1 at the moment for the case studies. As a result, the generated multiplicative factor for perturbation is homogenous. This is to ensure a satisfying spread among the ensemble members. Otherwise, if no spatial correlation is assumed, one can hardly see any differences among the ensemble members in terms of the basin-TWS, since the randomly generated perturbation shall cancel out, being averaged to zero.

*Does GLDA in the present form allow for such a parallelization? Of yes, why wasn't it used?*

It is a pity that the pre-processing module of PyGLDA does not support a parallelization at the moment. Since the pre-processing is required to perform for only once, we have not given it a priority for parallelization. But this will be added in next update as suggested.

*Thus, the total number of parameters to be stored is 6 x 2142930 ?  (to be clarified)*

It is indeed 11*2142930, since some of the vertical compartments has two layers in our GHM. For example, the soil water is defined as two layers because of the distinction of deep-rooted vegetation and shallow-rooted vegetation, see Figure 1b of the manuscript.

*How exactly is the phase defined? For instance, when is the TWS maximal if phase=0?*

In our computation, the phase is defined with the time when TWS is maximal, and the time is indicated by month. For example, phase=[0-1] indicates that maximum takes place at January.

**Comment 9:**

*I recommend to publish the manuscript provided that the authors update it in line with my remarks.*

**Response**: Thank you again for your valuable feedback. We hope these revisions address your concerns and improve the manuscript's clarity and impact.

---

## Author Response (AR2)

**Response to the Referee #3**

*Dear Dr. Fan and co-authors,*

*Following the long struggle to finish the review phase and acknowledging the very positive prior reviews, I undertook the final review of this manuscript. To acknowledge my expertise (and its lack): I am familiar with GRACE and global gravity measurements, and have done some work in developing fast/simplified global hydrological models. I am not an expert in the model at hand and have little/no direct experience in data assimilation (though I am familiar with it on a conceptual and operational level).*

*The manuscript is very well written, represents a significant advance in our modeling toolkit, and is in my opinion worthy of publication. Additionally, I think that the authors have done a good job of incorporating prior referee comments. I note here a few (largely minor) changes and suggestions:*

*Line 229: I was concerned initially about information leakage (especially using gravity for DA) across these boundaries. Lines 247-253 reassured me that this was being considered. No need to make changes, but consider whether it could be useful to address possible reader concerns around Line 229.*

**Response**: We appreciate your great efforts for the evaluation of our paper. Regarding the leakage issue, we have followed your suggestion and added a notice there, in line 228:

"The assumption of independency is indeed too ideal but will be addressed in the next section"

*Line 408. If I divide this through, I get 1.6 years model time per day of wall time. But was much of this wall time spent in the setup phase? If so, it could be interesting for readers to know the post-setup speed of the GLDA. It might also be more fair to your approach and highlight your advances.*

**Response**: Yes, actually over half of the wall time goes to the setup process, i.e., the data preparation, as evidenced from the Table 3 (time required for a patch) of the main text. Once the GLDA is fully set up, the implementation of DA is fairly fast. In line 407, we also gave an example that it took less than 15 minutes to perform a regional DA for Danube river basin. In the revised manuscript (see line 408), we add: 'but be aware that more than half of the time goes to the data preparation'.

*425-426. The heightened sensitivity on GRACE seems something to discuss, here or elsewhere. Does this mean that, once we include DA, it matters quite a bit less which base model we are using? I imagine this is a follow-up question for a later paper.*

**Response**: Thank you for sharing the insight. In fact, DA generally adopts the spatial details of the base model, but the magnitude is tuned towards the gravity measurements/observations. In addition, be aware that the comparison of main text is based on the upscaled model and DA output

(from 0.1 degree to 0.5 degree to be compatible with GRACE), so that the spatial details <0.5 degree are invisible, which underestimate the contribution of base model. Our view is that the base model is still fairly relevant for DA, particularly in determining the spatial details of the water variability.

*478-479. Because we cannot tell the future, I wonder about a good way to note next steps without necessarily promising that they will be done.*
**Response**: Thank you very much for the suggestion. In the revision (see line 472-477), we have tuned the tone and indicated that these are potential future extension instead of actions that must be done.

*480. Under the MIT license*
**Response**: Corrected as suggested.

*484-485. The policies of GMD would promote use of a permanent repository for model results rather than a Dropbox folder, which could be changed / removed over time. Could you publish the model implementation in a permanent repository? If space is a problem, then please be in contact.*
**Response**: In fact, our code and data have been collected together and uploaded to a permanent data repository with a DOI (Zenodo, https://zenodo.org/records/12206756), instead of a Dropbox folder. Please have a check and this should be complying with the GMD policy.

The Dropbox mentioned in the Data Availability section is indeed where the original model (in Matlab) is available. However, our PyGLDA uses a latest Python version of the model, which has already been uploaded to Zenodo repository.

*498. Could you define "f" closer to this and/or remove it from "Theta" if not needed?*
**Response**: Thank you for the careful review. To improve the manuscript, we had a further proofreading of the main text including the Appendix section. Some minor changes related to the language use and formulas have been made in the revised manuscript.